# Tipping interactions and cascades on multimillennial time scales in a model of reduced complexity

Victor Couplet<sup>1,2</sup> and Michel Crucifix<sup>1</sup>

**Correspondence:** Victor Couplet (victor.couplet@vub.be)

Abstract. A tipping cascade refers to a sequence of tipping events in the Earth system, where transitions in one subsystem can trigger subsequent transitions in other subsystems. These cascades represent a significant concern for the future, as the tipping of a single element could induce the tipping of interconnected elements that would not have otherwise crossed their thresholds. This chain reaction could lead to substantial and potentially irreversible changes in the Earth's system, even under low-emission scenarios. However, tipping cascades, particularly those involving ice sheets, may unfold over millennial timescales and are therefore rarely captured in state-of-the-art Earth system models, which typically run only until the end of the 21st century. In this study, we extend the simple climate model SURFER v3.0 to incorporate a network of interacting tipping elements and other nonlinear components. Using this extended model, we systematically investigate the occurrence of tipping events and cascades over multi-millennial timescales and under a range of realistic emission scenarios. We show that interactions among tipping elements generally increase their tipping risks, consistent with findings from previous studies. Furthermore, our results suggest that meeting the Paris Agreement target of limiting warming below 2 °C could lower the risk of observing tipping events and cascades by roughly an order of magnitude compared to current-policy pathways, underscoring the urgency of stronger climate action.

## 1 Introduction

A tipping element is a component of the Earth system that can undergo a qualitative state change once a critical forcing threshold, or tipping point, is crossed. This transition, driven by one or more amplifying feedback loops, may be abrupt and/or irreversible (Lenton et al., 2008; Canadell et al., 2021; Lenton et al., 2023). Examples of suspected tipping elements include the Greenland and West Antarctic ice sheets (GRIS and WAIS), the Amazon rainforest (AMAZ), and the Atlantic Meridional Overturning Circulation (AMOC). There is evidence of past tipping events in paleoclimate records (Brovkin et al., 2021; Boers et al., 2022), and increasing concern that similar events may occur in the future as a result of global warming (McNeall et al., 2011; IPCC, 2021; Armstrong McKay et al., 2022; Wang et al., 2023; Lenton et al., 2023). For instance, parts of the Amazon rainforest, the Greenland ice sheet, and the AMOC are exhibiting critical slowing down, a sign of declining stability (Dakos et al., 2008; Scheffer et al., 2009; Boulton et al., 2022; Boers, 2021; Ditlevsen and Ditlevsen, 2023; Boers and Rypdal, 2021).

<sup>&</sup>lt;sup>1</sup>Earth and Life Institute, UCLouvain, Louvain-la-Neuve, Belgium

<sup>&</sup>lt;sup>2</sup>Department of Water and Climate, Vrije Universiteit Brussel, Brussels, Belgium

50

Recent studies have even projected a 50% or higher chance of AMOC collapse within this century (Ditlevsen and Ditlevsen, 2023; van Westen et al., 2024b), though such predictions are debated (Ben-Yami et al., 2024).

Tipping elements, as parts of the Earth system, are linked together through oceanic and atmospheric circulation systems and can interact across scales in time and space (Rocha et al., 2018; Wunderling et al., 2021; Kriegler et al., 2009). A recent review of known interactions between tipping elements found that most are destabilizing (Wunderling et al., 2024). This implies that the tipping of one element could facilitate—or even trigger—the tipping of others, potentially resulting in a tipping cascade and leading to substantial, possibly irreversible changes in the Earth system. Some tipping elements also influence global mean temperature through albedo changes or the release of greenhouse gases, further destabilising other elements. Despite growing evidence and concern about these interactions, their strength and net influence on the risk of individual tipping events or cascades remain highly uncertain, partly because substantial uncertainties persist even when elements are studied in isolation. In this study, we address these uncertainties and explore the potential for tipping events and cascades using SURFER v3.0, a reduced-complexity Earth system model (Martínez Montero et al., 2022; Couplet et al., 2025).

Despite their simplicity, reduced-complexity and conceptual models offer several advantages for studying tipping interactions and cascades. They have been used to establish generic properties of tipping cascades (Dekker et al., 2018), or to study the dynamics emerging from interactions between specific elements such as the AMOC and ENSO (Dekker et al., 2018), sea ice and AMOC (Lohmann et al., 2021), or ice sheets and AMOC (Sinet et al., 2023; Klose et al., 2024). Their computational efficiency enables simulations over several thousand of years, required to capture the long-term response of slow components such as ice sheets. It also supports large simulation ensembles across a wide range of forcing scenarios and parameters, providing a practical framework to sample uncertainties. For instance, Wunderling et al. (2021) performed over a million simulations with a conceptual model of four tipping elements (GRIS, WAIS, AMOC, AMAZ) and found that interactions tend to destabilize the global system, increasing the likelihood of tipping events and cascades even under warming levels below 2°C. Their results also suggest that ice sheets often initiate tipping cascades, with the AMOC frequently acting as a mediator. However, in their simulations, the global mean temperature is kept constant at different warming levels, rather than allowed to evolve dynamically in time. This places a strong emphasis on the critical temperatures at which tipping elements might tip, while overlooking the importance of their internal dynamics time scale. Follow-up studies using the same model but non-stationary temperature forcings showed that the risk of tipping increases with peak temperature, convergence temperature, and the time needed to reach the convergence temperature (Wunderling et al., 2023; Möller et al., 2024). Deutloff et al. (2025) used a different conceptual model to assess how CO2 emissions from Amazon rainforest and permafrost collapse could influence the probability of other tipping points being crossed by 2500. They found a positive but relatively small effect, especially compared to scenario dependence.

Here, we "ride the wave" of reduced-complexity models and study tipping dynamics using SURFER v3.0, which we extend with parameterisations for several tipping elements. Building on the previously cited studies, we include in our model both interactions between elements and their feedbacks on global mean temperature, while also accounting for uncertainties through a Monte Carlo ensemble of simulations spanning several thousand years. This work aims to answer the following questions:

60

- Considering realistic emission scenarios resulting in non-stationary temperatures, how do the intrinsic time scales of tipping elements affect their tipping behaviour?
- To what extent do interactions and feedbacks on global mean temperature increase the risk of observing tipping events?
  - What are the cascades that could happen in the future for scenarios that stay within or close to the 2 °C Paris range?
  - Can we identify specific roles for the elements involved in a tipping cascade? Specifically, are the Greenland and West Antarctic ice sheets really the main initiators of cascades, despite having a slow response time scale?

In section 2 we describe our model, how we implement the tipping elements, and the overall experimental setup. Section 3 describes the results of our experiments. We focus on tipping elements individually in section 3.1, and their increased risk of tipping when interactions are taken into account. We then focus on tipping cascades in section 3.2. In section 4, we discuss our results and highlight the advantages and limitations of our approach.

#### 2 Methods

## 2.1 The SURFER model of reduced complexity

SURFER v3.0 is a simple Earth system model that reliably estimates CO<sub>2</sub> concentrations, global mean temperatures, sealevel rise, and many ocean acidification metrics in response to anthropogenic greenhouse gas emissions on timescales from decades to millions of years (Couplet et al., 2025). Version 3.0 consists of 17 differential equations: eleven describe carbon exchanges between reservoirs (atmosphere, land, upper, intermediate, and deep ocean layers, and deep-sea sediments); three describe temperature anomalies in the ocean layers; two describe the volumes of the Greenland and Antarctic ice sheets; and one describes sea-level rise from glaciers.

In the extended version used here, we keep the Greenland ice sheet (GRIS), and we split the Antarctic ice sheet into the West Antarctic ice sheet (WAIS), the East Antarctic subglacial basins (EASB), and the remainder of the East Antarctic ice sheet (EAIS). We also add parametrisations for the Atlantic meridional overturning circulation (AMOC), the Amazon rainforest (AMAZ), boreal permafrost (PERM), and Arctic sea-ice (ASI). Since our model does not feature an explicit seasonal cycle, we consider mean annual Arctic sea-ice, not distinguishing between winter sea ice and summer sea ice. All these systems, important for climate, were classified as global core tipping elements in Armstrong McKay et al. (2022).

Mountain glaciers (GLCR), which are also included in this extended version, are classified as regional tipping elements in Armstrong McKay et al. (2022). However, when aggregated globally, they tend to respond fairly linearly to temperature changes (Rounce et al., 2023; Zekollari et al., 2024). The classification of Arctic sea-ice and boreal permafrost as tipping elements is also debated. In the 2023 Global Report on Tipping points, they were not classified as tipping systems (Lenton et al., 2023). Nevertheless, we include them in this study because of their potential to produce strong feedbacks on temperature, and because of their close coupling with other tipping elements (Wunderling et al., 2024). Since neither exhibits signs of bistability, we refer to them here as *monostable elements* (MEs). By contrast, we refer to GRIS, WAIS, EASB, EAIS, AMOC, and AMAZ

Figure 1. Conceptual diagram of SURFER with the inclusion of interacting tipping elements and their feedbacks on climate.

as proper *tipping elements*. When discussing all elements collectively, we use the term *nonlinear elements* (NEs) or simply *elements*. Any reference to an element "tipping" implicitly refers specifically to a tipping element.

The following sections describe the parametrisations we use for the elements and their interactions, and the implementation of the elements feedbacks on temperature, either through albedo changes or additional greenhouse gas. The different components of the model and the interactions between those components are schematically summarised in Figure 1.

#### 2.2 Representation of tipping elements and other nonlinear components

The evolution of the sea-level rise contribution from glaciers is the same as in previous SURFER versions and is given by the equation

$$\frac{\mathrm{d}S_{\mathrm{gl}}}{\mathrm{d}t} = \frac{1}{\tau_{\mathrm{el}}} \left( S_{\mathrm{gleq}}(\delta T_{\mathrm{U}}) - S_{\mathrm{gl}} \right),\tag{1}$$

with

$$S_{\text{gleq}}(\delta T_{\text{U}}) = S_{\text{gl pot}} \tanh\left(\frac{\delta T_{\text{U}}}{\zeta}\right).$$
 (2)

Here  $\tau_{\rm gl}$  is a relaxation timescale,  $S_{\rm gl\ pot}$  is the potential sea level rise due to mountain glaciers, and  $\zeta$  is a sensitivity coefficient. The state evolution of the other tipping elements and nonlinear components is modelled by

$$\frac{\mathrm{d}x_i}{\mathrm{d}t} = f(x_i, q_i) = \left[\underbrace{-x_i^3 + a_{2,i}x_i^2 + a_{1,i}x_i + c_{0,i}}_{\text{H}_i} + c_{1,i}\delta q_i\right] \mu(x_i, H_i), \tag{3}$$

with  $i \in \{GRIS, WAIS, EASB, EAIS, ASI, AMOC, AMAZ, PERM\}$ . More specifically,  $x_{GRIS}$ ,  $x_{WAIS}$ ,  $x_{EASB}$ , and  $x_{EAIS}$  represent the fraction of ice volumes compared to pre-industrial levels;  $x_{ASI}$  represent the fraction of mean annual Arctic sea-ice

surface compared to pre-industrial values;  $x_{\rm AMOC}$  represents the fraction of AMOC strength compared to pre-industrial values; and  $x_{\rm AMAZ}$  and  $x_{\rm PERM}$  represent the fraction of carbon stored in the Amazon Rainforest and permafrost respectively, compared to pre-industrial levels. The first group of terms in equation 3 represents the internal dynamics of the element. Since it is a cubic polynomial of the variable  $x_i$ , it allows the elements to have 1, 2, or 3 stable states depending on  $\delta q_i$ , which represents the forcing anomaly relative to pre-industrial times ( $\delta q_i(t_{\rm PI})=0$ ). The factor  $\mu(x_i,H_i)$  is an inverse time scale and is given by

$$\mu(x_i, H_i) = \begin{cases} 1/\tau_i^+ & \text{if } H_i > 0 \text{ and } 0 < x_i < 1, \\ 1/\tau_i^- & \text{if } H_i < 0 \text{ and } 0 < x_i < 1, \\ 0 & \text{if } x_i \le 0 \text{ or } x_i \ge 1. \end{cases}$$
(4)

The dependence of  $\mu$  on the sign of H allows the representation of the asymmetric character of processes driving the evolution of some elements. For instance, ice sheets tend to melt faster than they form. The third case in Eq.4 constrains  $x_i$  to remain between 0 and 1, with  $x_i=0$  indicating a completely collapsed state and  $x_i=1$  indicating the pre-industrial state. While the first constraint is physical, the second one is introduced here to facilitate the analysis and the parametrisation of the interactions between elements (see section 2.4.2). Furthermore, in the numerical implementation of the model, the function  $\mu$  is smoothed out such that the right-hand side of Eq. 3 is differentiable everywhere.

The stability structure induced by Eq. 3 depends on the constant parameters  $(a_2, a_1, c_1, c_0)$ . When  $a_2^2 + 3a_1 > 0$ , f(x,q) has 1, 2 or 3 real roots depending on  $\delta q$ , and we have a double-fold structure: there are three equilibrium branches, two sable,  $x_+^s(\delta q)$  and  $x_-^s(\delta q)$ , and one unstable  $x^u(\delta q)$ , that meet at critical points  $(q_+, x_+)$  and  $(q_-, x_-)$  in two fold bifurcations (see Figure 2a). For  $q_- 

with

$$G = \frac{q_+ + q_- + 2\sqrt{q_+ q_-}}{q_+ - q_-} \,. \tag{10}$$

When  $a_2^2 + 3a_1 

Figure 2. Examples of steady state structures produced by Equation 3 for the elements. (a) For  $a_2^2 + 3a_1 > 0$ , there is a bistable double-fold stability structure. Parameters  $a_2, a_1, c_1, c_0$  can be expressed as the coordinates of the two bifurcation points  $(q_+, x_+)$  and  $(q_-, x_-)$ . (b) For  $a_2^2 + 3a_1 

190

195

**Table 1.** Parameters for the tipping elements and other nonlinear monostable elements. For  $T_+$  and  $\tau_-$ , parameters are sampled uniformly in the ranges based on the literature (see section 2.4). Other parameters are fixed. For the AMOC, the AMAZ and ASI,  $\tau_+$  is set equal to the sampled value for  $\tau_-$ . For the Arctic sea ice (ASI) temperature threshold  $T_I$ , we use the range given in Armstrong McKay et al. (2022) for the winter sea ice, as it is responsible for the nonlinear behaviour. Similarly, for our permafrost temperature threshold  $T_I$ , we take the values given in Armstrong McKay et al. (2022) for the boreal permafrost collapse (PFTP).

| Tipping  | $T_{+}$     | $T_{-}$ | $x_{+}$            | $x_{-}$ | au          | $	au_+$    | $ar{F}$                       | $\bar{E}^{\mathrm{CO}_2}$ | $\bar{E}^{\mathrm{CH_4}}$ |
|----------|-------------|---------|--------------------|---------|-------------|------------|-------------------------------|---------------------------|---------------------------|
| elements | (°C)        | (°C)    | adim               | adim    | (years)     | (years)    | $(\mathrm{W}\mathrm{m}^{-2})$ | (PgC)                     | (PgC)                     |
| GRIS     | [0.8, 3.0]  | 0.0     | 0.75               | 0.25    | [306, 1981] | 5500       | 0.169                         | 0                         | 0                         |
| WAIS     | [1.0, 3.0]  | 0.0     | 0.75               | 0.25    | [153, 1717] | 5500       | 0.065                         | 0                         | 0                         |
| EASB     | [2.0, 6.0]  | 0.0     | 0.75               | 0.25    | [88, 839]   | 5500       | 0.065                         | 0                         | 0                         |
| EAIS     | [5.0, 10.0] | 0.0     | 0.75               | 0.25    | [470, 741]  | 5500       | 0.78                          | 0                         | 0                         |
| AMOC     | [1.4, 8.0]  | 0.0     | 0.75               | 0.25    | [2.2, 21]   | $=	au_{-}$ | 0                             | 0                         | 0                         |
| AMAZ     | [2.0, 6.0]  | 0.0     | 0.75               | 0.25    | [7.9,15]    | =	au       | 0                             | 75                        | 0                         |
| Other    | $T_{I}$     | $x_I$   | $d_I$              |         |             |            |                               |                           |                           |
| elements | (°C)        | adim    | $(^{\circ}C^{-1})$ |         |             |            |                               |                           |                           |
| ASI      | [4.5, 8.7]  | 0.5     | -0.50              |         | [1.9, 8.4]  | $=	au_{-}$ | 0                             | 0                         | 0                         |
| PERM     | [3.0, 6.0]  | 0.5     | -0.25              |         | [5.3, 76]   | 3000       | 0                             | 300                       | 7.5                       |

elements,  $T_{-}$  and  $x_{-}$  or  $x_{I}$  and  $d_{I}$ , and  $\tau_{+}$  are fixed. This is also the case for the parameters  $\bar{F}$ ,  $\bar{E}^{CO_{2}}$ , and  $\bar{E}^{CH_{4}}$  that regulate the feedbacks of the elements on the temperature (see section 2.4.3). Table 1 summarises the values given to the parameters for the elements, and more details are provided in Section 2.4.

For each parameter set of our ensemble, we run the extended SURFER model from 1850 CE until the year 100000 CE for a family of 100 emission scenarios. This forms a single experiment with  $1000 \times 100 = 100000$  model runs. The emission scenarios start in 1850 CE and follow historical CO<sub>2</sub> emissions until 2024 CE. After that, between 50 and 5000 PgC of CO<sub>2</sub> are released in the atmosphere following a logistic equation, with each successive scenario adding 50 PgC. Methane emissions are the same in all scenarios: they follow historical emissions from 1850 CE until 1995 CE, then follow the SSP1-2.6 scenario. More details about the construction of these scenarios are in Appendix A. For the same parameters and without including the feedbacks of the elements on temperature, the temperature difference in our model between two successive scenarios at any given time is lower than  $0.1^{\circ}$ C (for an equilibrium climate sensitivity (ECS) of  $3.5^{\circ}$ C). Throughout this paper, we will refer to scenarios by their CO<sub>2</sub> cumulative emissions value after 2024, rather than their total cumulative emission value since 1850.

We perform several experiments for different model settings. The two main experiments are the *decoupled* and *coupled* experiments. For the decoupled experiment, we switch off the interactions between the elements (d=0) and their feedbacks on temperature  $(\bar{F}_i=0, \bar{E}_i^{\text{CO}_2}=0, \bar{E}_i^{\text{CH}_4}=0)$ . In the coupled experiment, they are taken into account. We also perform an *interactions-only* experiment where feedbacks of elements on temperature are turned off, and a *feedback-only* experiment where

Table 2. Literature-based temperature thresholds and transition time scales for tipping elements, as given in Armstrong McKay et al. (2022).

| TI.      | Critic | al tempe | rature (°C) | Transition timescale (years) |       |      | Max impact |
|----------|--------|----------|-------------|------------------------------|-------|------|------------|
| Elements | Min    | Max      | Best        | Min                          | Max   | Best |            |
| GRIS     | 0.8    | 3.0      | 1.5         | 1k                           | 15k   | 10k  | 0.13°C     |
| WAIS     | 1.0    | 3.0      | 1.5         | 500                          | 13k   | 2k   | 0.05°C     |
| EASB     | 2.0    | 6.0      | 3.0         | 500                          | 10k   | 2k   | 0.05°C     |
| EAIS     | 5.0    | 10.0     | 7.5         | $10 \text{k}^*$              | $?^*$ | ?    | 0.60°C     |
| AMOC     | 1.4    | 8.0      | 4.0         | 15                           | 300   | 50   | -0.60°C    |
| AMAZ     | 2.0    | 6.0      | 3.5         | 50                           | 200   | 100  | 75 PgC     |
| AWSI     | 4.5    | 8.7      | 6.3         | 10                           | 100   | 20   | 0.60°C     |
| PERM     | 3.0    | 6.0      | 4.0         | 10                           | 300   | 50   | ?**        |

<sup>\*</sup>For our calibration procedure of  $\tau_-$  (see section 2.4.1), we use minimum estimate of 5 kyr instead of 10 kyr for the transition time scale of the East Antarctic ice sheet (EAIS), and a maximum estimate of 15 kyr. This provides a better estimation of committed sea level rise. \*\*Armstrong McKay et al. (2022) divides the permafrost into three categories and doesn't provide a single number for the maximum impact of permafrost as a whole.

interactions between elements are turned off. In Appendix D, we provide results for experiments with a higher equilibrium climate sensitivity, with stronger interactions between tipping elements, and without considering the nonlinear monostable elements ASI and PERM. In total, we performed  $1000 \times 100 \times 8 = 800000$  model runs, representing together around 80 billion years of simulation.

#### 200 2.4 Calibration

## 2.4.1 Critical temperatures and internal timescales

For each tipping element, the temperature threshold  $T_+$  is sampled within the intervals provided in Table 2, based on the review by Armstrong McKay et al. (2022). These intervals align closely with values in the more recent Global Tipping Points Report (Lenton et al., 2023). There is limited information on  $x_+$  and the "back-tipping" point  $(T_-, x_-)$  in the literature, including in specific assessments of tipping elements. To streamline our analysis, we set  $x_+ = 0.75$  and  $T_- = 0$  for all tipping elements. The corresponding value of  $x_-$  is then determined by equations 9-10, resulting in  $x_- = 0.25$  for all elements, independent of the value given to  $T_+$ .

Both  $x_+ = 0.75$  and  $x_- = 0.25$  are physically plausible. The value  $T_- = 0$  may be less realistic for certain tipping elements. For example, the hysteresis experiment in Garbe et al. (2020) would impose  $T_{-,EAIS} > 5$ °C. As our focus here is primarily on the "forward tipping points", i.e., transitions from the baseline upper branch regime to the tipped lower branch regime (see figure 2),  $T_-$  does not affect our results and we maintain it to 0 for mathematical simplicity.

220

225

For Arctic sea ice (ASI) and permafrost (PERM), which are here considered as nonlinear but monostable elements ( $a_2^2 + 3a_1 > 0$ ), we fix  $x_I = 0.5$ , and we sample  $T_I$  within the intervals provided in Table 1, based on the review by Armstrong McKay et al. (2022), as for the tipping elements. For Arctic sea ice, we set  $d_I = -0.5^{\circ}\text{C}^{-1}$ , and for permafrost we set  $d_I = -0.25^{\circ}\text{C}^{-1}$ , which, together with the set  $T_I$  range, results in a slightly nonlinear dependence of its equilibrium state on temperature.

The parameters  $\tau_-$  and  $\tau_+$  in equations 3-4 describe the internal dynamics of elements. In particular,  $\tau_-$  determines the transition time of an element from its baseline state to its collapsed state. Estimates of transition times by Armstrong McKay et al. (2022) are reproduced in Table 2. We sample  $\tau_{-,i}$  in a range  $\left[\tau_{-,i}^{\min},\tau_{-,i}^{\max}\right]$  chosen such that if the element i tips, its transition time falls within (or close to) the range given by Armstrong McKay et al. (2022). The ranges  $\left[\tau_{-,i}^{\min},\tau_{-,i}^{\max}\right]$  are presented in Table 1, and more details on how we determine them are provided in Appendix C.

For the AMOC, Amazon rainforest (AMAZ), and Arctic sea ice (ASI), we set  $\tau_+$  equal to the sampled value for  $\tau_-$ , indicating a symmetric timescale. For the ice sheets (GRIS, WAIS, EASB, and EAIS), we set  $\tau_+ = 5500$  years, a much higher value than  $\tau_-$ , to capture the asymmetry between the faster melting and slower freezing processes. For permafrost, while the frozen soil area is considered reversible with decreasing temperatures, the burial of organic carbon is irreversible on short timescales (Lenton et al., 2023). This asymmetry between rapid carbon release upon thaw and the much slower carbon capture when soil refreezes is represented in our model by defining  $x_{\rm PERM}$  as the releasable carbon content in permafrost (rather than permafrost area) and setting  $\tau_+ = 3000$  years, a value much higher than  $\tau_-$  (see Table 1).

#### 2.4.2 Interactions between the elements

For a time evolution described by Equation 3, an element i will tip from the upper branch  $x_s^+(\delta q)$  to the lower branch  $x_s^-(\delta q)$  if the forcing  $\delta q_i = \delta T_U + d \cdot \sum_j \epsilon_{ij} \delta L_j q_{*,i}$  exceeds the critical value  $q_{*,i} = T_{+,i}$  for a long enough time. In the case of monostable elements, exceeding the critical value  $q_{*,i} = T_{I,i}$  drives the system past its inflection point. If we neglect warming  $(\delta T_U = 0)$  and consider an element i forced by a unique quantity  $\delta L_j$  ( $\delta L_{k \neq j} = 0$ ), we have  $\delta q_i = \epsilon_{ij} \delta L_j q_+$  and the element i will tip (or go past its inflection point) for  $\epsilon_{ij} \delta L_j > 1$ . In other words, the critical value  $L_j^+$  of the quantity  $\delta L_j$  is given by

$$L_j^+ = \frac{1}{\epsilon_{ij}} \,, \tag{18}$$

implying that if we can derive it from observations or from appropriate model experiments, we can set  $\epsilon_{ij} = \frac{1}{L_i^+}$ .

Consider the AMOC. As temperatures increase, the changes in the evaporation-precipitation budget over the Atlantic region affect the salinity of surface waters. This may cause a slowdown of the AMOC, and eventually a collapse (Rahmstorf, 2024). These processes are represented by the forcing term  $c_1\delta T_U$ . The critical temperature for AMOC tipping is estimated to lie between 1.4°C and 8°C of warming (Armstrong McKay et al., 2022). We use this knowledge to set  $q_+ = T_+$ . The AMOC may also be impacted by the freshwater flux  $F_{\rm GRIS}$  coming from the melting Greenland ice sheet. This flux can be expressed in sverdrups as

$$F_{\text{GRIS}} = -\frac{S_{\text{pot,GRIS}} A_O}{3600 \cdot 24 \cdot 365 \cdot 10^6} \cdot \dot{x}_{\text{GRIS}}, \tag{19}$$

where  $S_{\text{pot,GRIS}}$  is the sea level rise potential of the Greenland ice-sheet and  $A_O$  is the ocean surface area ( $S_{\text{pot,GRIS}} = 7$  m and  $A_O \sim 360 \times 10^6 \text{ km}^2$ ). The impact of this meltwater flux on the AMOC can then be included in a coupling term  $\epsilon_{\text{AMOC,GRIS}} F_{\text{GRIS}}$ . Whether meltwater fluxes from Greenland could be sufficient to tip AMOC is not certain (Weijer et al., 2019). According to an early EMIC inter-comparison project, AMOC could tip with a hosing flux from 0.1 sv to 0.5 sv (Rahmstorf et al., 2005). Recently, van Westen et al. (2024a) found a tipping point at 0.5-0.6 Sv in CESM but the model has salinity biases, and the tipping point may occur for a smaller forcing. Here, based on Rahmstorf et al. (2005), we assume that the critical meltwater flux  $F_{\text{GRIS}}^+$  lies in the range [0.1, 0.5] Sv and we sample  $\epsilon_{\text{AMOC,GRIS}}$  in  $\left[\frac{1}{0.5}, \frac{1}{0.1}\right] = [2, 10]$ .

Meltwater from the west-Antarctic ice-sheet (WAIS) could also impact the AMOC, and we model it through the coupling term  $\epsilon_{\text{AMOC,WAIS}}F_{\text{WAIS}}$ , with

$$F_{\text{WAIS}} = -\frac{S_{\text{pot,WAIS}} A_O}{3600 \cdot 24 \cdot 365 \cdot 10^6} \cdot \dot{x}_{\text{WAIS}}, \qquad (20)$$

where  $S_{\text{pot,WAIS}}$  is the sea level rise potential of the West Antarctic ice-sheet ( $S_{\text{pot,GRIS}} = 4 \text{ m}$ ). In this case, it is less clear what the critical flux  $F_{\text{WAIS}}^+$  should be. Wunderling et al. (2024) considers the WAIS  $\rightarrow$  AMOC interaction as weak to moderate, while it considers the GRIS  $\rightarrow$  AMOC interaction strong. Based on this, we decide that the maximum value that  $\epsilon_{\text{AMOC,WAIS}}$  can take is 2/3 of the maximum value that  $\epsilon_{\text{AMOC,GRIS}}$  can take. With this rule, for a given meltwater flux, the maximal impact of WAIS on AMOC is smaller than the maximal impact of GRIS, which satisfies Wunderling et al. (2024). The sign of the WAIS  $\rightarrow$  AMOC interaction is not clear either. A melting WAIS could either stabilise or destabilise the AMOC because there are competing effects. We thus sample  $\epsilon_{\text{AMOC,WAIS}}$  in the range  $\left[-\frac{2}{3} \cdot 10, \frac{2}{3} \cdot 10\right]$ . A negative  $\epsilon_{\text{AMOC,WAIS}}$  implies a stabilising interaction.

The other interactions are less well known. Choosing appropriate  $\delta L_j$  and  $L_j^+$  (or  $\epsilon_{ij}$ ) is then harder. We adopt simple linear coupling rules, as in Wunderling et al. (2021). We set  $\delta L_j = (1 - x_j)$  such that the forcing from element j on element i is zero at pre-industrial times and maximal when j has completely collapsed ( $x_j = 0$ ). Then, for all element pairs (i,j) except (AMOC,GRIS) and (AMOC,WAIS), we sample  $\epsilon_{ij}$  in intervals based on the assessed strength and signs of the interactions by Wunderling et al. (2024), following the rule given in Figure 3. If no interaction is identified from element j to i, we set  $\epsilon_{ij} = 0$ .

## 2.4.3 Feedbacks on climate

Tipping elements, or other nonlinear elements, may impact the global mean temperature. This can happen for example through additional greenhouse gas emissions for the biospheric elements (AMAZ, PERM) or through changes in albedo for the cryospheric elements (ASI, GRIS, WAIS, EASB, EAIS, and mountain glaciers). We modify the climate and carbon components of SURFER v3.0 to include these processes. The evolution of temperature anomaly in the surface ocean layer,  $\delta T_U$ , which is considered to be in thermal equilibrium with the atmosphere, is dictated by

$$c_{\text{vol}} h_{\text{U}} \frac{d\delta T_{\text{U}}}{dt} = F\left(M_{\text{A}}, M_{\text{A}}^{\text{CH}_4}, I\right) - \beta \delta T_{\text{U}} + \sum_{i} F_{\text{NE}, i}(x_i) - \gamma_{\text{U} \to \text{I}} \left(\delta T_{\text{U}} - \delta T_{\text{I}}\right). \tag{21}$$

Figure 3. Assessment and parametrisation of the interactions between the elements. (a) Interactions between tipping elements and other nonlinear components. Colours give the sign of the interaction (Stabilising, Destabilising, Competing effects or unclear), and the letter gives the assessed strength (Strong, Moderate, Weak, or unclear). Adapted from the review on climate tipping point interactions from Wunderling et al. (2024), and more details on specific interactions and their physical mechanisms can be found therein. We added EASB, and we have modelled its interactions with other elements similar to WAIS, except for EASB $\rightarrow$ AMOC, which is set to zero. This choice reflects the high uncertainty in both the sign and strength of the WAIS $\rightarrow$ AMOC interaction, making an EASB $\rightarrow$ AMOC link quite uncertain. (b) Sampling ranges for the parameters  $\epsilon_{ij}$  that define interactions in the model, as functions of the assessed sign and strength of the interactions by Wunderling et al. (2024). For the interaction from GRIS to AMOC and from WAIS to AMOC,  $\epsilon_{ij}$  is sampled in different ranges than the ones provided here, respectively [2, 10] and [-20/3, 20/3].

The term  $F(M_A, M_A^{CH_4}, I)$  is the radiative forcing due to greenhouse gases and aerosols. Its expression is given by

$$F\left(M_{\rm A}, M_{\rm A}^{\rm CH_4}, I\right) = F_{2\times} \log_2\left(\frac{M_{\rm A}}{M_{\rm A}\left(t_{\rm PI}\right)}\right) + \alpha_{\rm CH_4} \sqrt{M_{\rm A}^{\rm CH_4} - M_{\rm A}^{\rm CH_4}(t_{\rm PI})} - \alpha_{\rm SO_2} \exp\left(-\left(\beta_{\rm SO_2}/I\right)^{\gamma \rm SO_2}\right). \tag{22}$$

The first two terms describe the contribution of  $CO_2$  and methane to an increased greenhouse effect. The third term corresponds to solar radiation modification in the form of  $SO_2$  injections. In this study, they are always set to zero. The term  $\gamma_{U \to I} (\delta T_U - \delta T_I)$  models heat exchanges with the intermediate ocean layer. The climate feedback parameter  $\beta$  represents the feedbacks of fast processes (centennial time scale) and is related to the equilibrium climate sensitivity (ECS, in °C)

$$ECS = \frac{F_{2\times}}{\beta} \tag{23}$$

with  $F_{2\times}$  the radiative forcing corresponding to a doubling of CO<sub>2</sub>. As in previous SURFER versions, we set  $F_{2\times} = 3.9 \mathrm{Wm}^{-2}$  and  $\beta = 1.1143 \mathrm{Wm}^{-2} \, \mathrm{C}^{-1}$ , such that we have an ECS equal to  $3.5 \, \mathrm{C}$ .

The term  $\sum_{i} F_{NE,i}(x_i)$  represents the slower feedbacks from the tipping and other nonlinear elements. It is the only new term in equation 21 compared to SURFER v3.0. For GRIS, WAIS, EASB, and EAIS, we set

$$F_{\text{NE},i}(x_i) = \bar{F}_i(1 - x_i^{4/5}) \tag{24}$$

where  $\bar{F}_i$  is the top-of-atmosphere radiative anomaly resulting from the albedo change following the complete removal of element i. The 4/5 exponent converts ice sheet volume to ice sheet surface, based on the volume-area scaling law for ice-caps (Bahr et al., 2015). The parameters  $\bar{F}_i$  are calibrated such that the additional warming from a complete removal of the elements in SURFER matches the warming  $\Delta T_{\text{ref},i}$  given in Wunderling et al. (2020), would SURFER have an ECS of 3°C. In other words, we set  $\bar{F}_i = \beta_{\text{ref}} \Delta T_{\text{ref},i}$  with  $\beta_{\text{ref}} = \frac{\bar{F}_{2\times}}{3^{\circ}\text{C}}$ . This is done to be compatible with Wunderling et al. (2020), who obtained mean values of  $\Delta T_{\text{ref},i}$  using a model ensemble with an ECS range of 2.0 to 3.75°C. With the parameters  $\bar{F}_i$  set, the effective temperature impacts of element removal in SURFER will scale with ECS, which is set here to 3.5°C (in Appendix D, we perform experiments with an ECS equal to 5°C). The  $\Delta T_{\text{ref},i}$  values from Wunderling et al. (2020) are the ones given in Armstrong McKay et al. (2022) for the maximum impact of ice-sheet tipping elements, and are reproduced in table 2.

We assume that ASI, AMAZ, and PERM vary fast enough that their feedback on global mean temperature through albedo is already captured in the climate feedback parameter  $\beta$ . This is also the case for the impacts of AMAZ on temperatures through evapotranspiration and cloud formation.

AMAZ and PERM can additionally impact global mean temperature through emissions of  $CO_2$  and  $CH_4$ . The evolution of the atmospheric  $CO_2$  and  $CH_4$  content  $(M_A, M_A^{CH_4})$  is described by the following equations

$$\frac{dM_{\rm A}}{dt} = V + E_{\rm fossil}^{\rm CO_2} + E_{\rm land-use}^{\rm CO_2} + E_{\rm NE}^{\rm CO_2} - F_{\rm A \to U} - F_{\rm A \to L} + F_{\rm CH_4, ox} - E_{\rm natural}^{\rm CH_4} - F_{\rm weathering},$$
(25)

$$\frac{dM_{\rm A}^{\rm CH_4}}{dt} = E_{\rm fossil}^{\rm CH_4} + E_{\rm land-use}^{\rm CH_4} + E_{\rm NE}^{\rm CH_4} + E_{\rm natural}^{\rm CH_4} - F_{\rm CH_4,ox}, \tag{26}$$

The meaning and expression of each term is detailed in Couplet et al. (2025). Here, we added  $CO_2$  and  $CH_4$  emissions ( $E_{NE}^{CO_2}$  and  $E_{NE}^{CH_4}$ ) from nonlinear elements. They are treated similarly to land-use emissions, and are thus taken from the land reservoir ( $M_L$ ):

$$05 \quad \frac{dM_{\rm L}}{dt} = F_{\rm A \to L} - E_{\rm land-use}^{\rm CO_2} - E_{\rm land-use}^{\rm CH_4} - E_{\rm NE}^{\rm CO_2} - E_{\rm NE}^{\rm CH_4}, \tag{27}$$

$$\frac{dM_{\rm L}^*}{dt} = -E_{\rm land-use}^{\rm CO_2}(t) - E_{\rm NE}^{\rm CO_2} - E_{\rm NE}^{\rm CH_4}, \tag{28}$$

The variable  $M_L^*$  is an ad-hoc variable that is used in the definition of the atmosphere to land carbon flux  $F_{\rm A\to L}$  and which can be thought of as the amount of carbon that the land can hold for  $\delta T_U=0$ . Including  $E_{\rm NE}^{\rm CO_2}$  and  $E_{\rm NE}^{\rm CH_4}$  in equation 28 means that the carbon loss is irreversible until the permafrost and the Amazon rainforest regrow.

Emissions from nonlinear elements are given by

$$E_{\rm NE}^{\rm CO_2} = E_{\rm AMAZ}^{\rm CO_2} + E_{\rm PERM}^{\rm CO_2},$$
 (29)

$$E_{\rm NE}^{\rm CH_4} = E_{\rm PERM}^{\rm CH_4},\tag{30}$$

with

$$E_{\text{AMAZ}}^{\text{CO}_2} = -C_{\text{AMAZ}}^{\text{CO}_2} \cdot \dot{x}_{\text{AMAZ}}, \tag{31}$$

and


$$E_{\text{PERM}}^{\text{CO}_2} = \begin{cases} -C_{\text{PERM}}^{\text{CO}_2} \cdot \dot{x}_{\text{PERM}} & \text{if } \dot{x}_{\text{PERM}} < 0, \\ -\left(C_{\text{PERM}}^{\text{CO}_2} + C_{\text{PERM}}^{\text{CH}_4}\right) \cdot \dot{x}_{\text{PERM}} & \text{if } \dot{x}_{\text{PERM}} > 0, \end{cases}$$
(32)

$$E_{\text{PERM}}^{\text{CH}_4} = \begin{cases} -C_{\text{PERM}}^{\text{CH}_4} \cdot \dot{x}_{\text{PERM}} & \text{if } \dot{x}_{\text{PERM}} < 0, \\ 0 & \text{if } \dot{x}_{\text{PERM}} > 0. \end{cases}$$
(33)

For permafrost emissions, we differentiate thawing ( $\dot{x}_{PERM} < 0$ ) and formation ( $\dot{x}_{PERM} > 0$ ). Indeed, when it thaws, the organic carbon that it contains is released either as  $CO_2$  or as  $CH_4$ , but the organic carbon that slowly accumulates in the active layer during the warm seasons and that is permanently captured as the soil refreezes comes from photosynthesis and thus atmospheric  $CO_2$ .

The coefficients  $C_i^{\text{CO}_2/\text{CH}_4}$  represent the maximal amount of carbon releasable by element i as  $\text{CO}_2$  or  $\text{CH}_4$ . Indeed, we have

$$\int_{x_{i}=1}^{x_{i}=0} E_{i}^{\text{CO}_{2}/\text{CH}_{4}} dx_{i} = -C_{i}^{\text{CO}_{2}/\text{CH}_{4}} \int_{x_{i}=1}^{x_{i}=0} \dot{x}_{i} dx_{i} = C_{i}^{\text{CO}_{2}/\text{CH}_{4}}.$$
(34)

For AMAZ, we set  $C_{\rm AMAZ}^{\rm CO_2} = 75\,{\rm PgC}$  based on Armstrong McKay et al. (2022). For PERM, estimates of CO<sub>2</sub> and CH<sub>4</sub> emissions are not well constrained. It is generally considered that permafrost soils hold around 1035 PgC of carbon (Schuur et al., 2015) but not all this carbon is necessarily released to the atmosphere upon complete thaw. Here we set  $C_{\rm PERM}^{\rm CO_2} = 300\,{\rm PgC}$  and  $C_{\rm PERM}^{\rm CH_4} = 7.5\,{\rm PgC}$  meaning that around 2.4% of the carbon is emitted as CH<sub>4</sub>. This fraction has been estimated between 2.3% and 2.7% from expert reviews (Schuur et al., 2013, 2015) but could range from 1 to 12% Schuur et al. (2022), and overall there is low confidence in the relative roles of CO<sub>2</sub> and CH<sub>4</sub> (Canadell et al., 2021). The total amount of carbon releasable is 307.5 PgC, or 1370 PgCO<sub>2</sub>eq (using a 100 year global warming potential for CH<sub>4</sub> relative to CO<sub>2</sub> equal to 27). This is comparable to the value of 1330 PgCO<sub>2</sub>eq cited in the Global Tipping Points report for permafrost carbon release at 10°C warming (Lenton et al., 2023), at which point we can assume that all the carbon has been released.

For the AMOC, changes in the oceanic circulation and exchanges between the surface and deep waters could impact the overall heat and carbon exchanges with the atmosphere. For example, Boot et al. (2024), found in CESM2 and for the SSP5-8.5 scenario, an atmospheric pCO2 decrease of 1.3 ppm per sverdrup decrease in AMOC strength. Furthermore, since AMOC





causes a large inter-hemispheric transport of heat, a collapse would cool the North Atlantic and Northern hemisphere, and warm the Southern hemisphere. These initial cooling and warming responses are then amplified asymmetrically by sea-ice albedo feedbacks and other atmospheric feedbacks, which result in an overall decrease in global mean temperature in the first few centuries after the collapse (Laurian et al., 2009; Drijfhout, 2015b). This cooling could be compensated by simultaneous global warming (Drijfhout, 2015a; van Westen et al., 2024b). What happens on longer time scales is less certain. Drijfhout (2015b) shows that a reorganisation of radiative transfers between the ocean and atmosphere reduces the cooling and even leads to a slight warming, before returning to cooling around 700 years after the collapse. Capturing such a complex response is beyond the capabilities of SURFER v3.0, which only includes a 3-layer ocean model, does not include high latitude boxes, nor any horizontal resolution, and does not represent moisture and water vapor in the atmosphere. Consequently, we choose to neglect the global pCO<sub>2</sub> and temperature response of an AMOC collapse. Nevertheless, the local temperature effects of AMOC on other elements (ASI, PERM, GRIS, ...), as well as other climate impacts such as changes in precipitation patterns over the Amazon rainforest, are taken into account through the coupling terms in Equation 3.

#### 3 Results

At the end of every model run, we assign every tipping element a *tipping status*, which is either *tipped* or *not tipped*. An element i is assigned the status *tipped* if there exists a time  $t \in [1850, 100000]$  such that  $x_i(t) 




## **Individual tipping elements**

We first examine the factors that govern the tipping behaviour of an element in the decoupled experiment. This will provide a reference to investigate the roles of interactions and feedbacks. With  $T_-$  and  $x_+$  set to the same value for all tipping elements, we focus on the effects associated with their critical temperature  $T_+$  and internal timescales  $\tau_-$  (and  $\tau_+$ ).

In the decoupled experiment, each element is forced by temperature only ( $\delta q = \delta T_{\rm U}$ ), determined by the emission scenario. Under a constant temperature forcing, an element tips if  $\delta q > q_+$  or  $\delta T_{\rm U} > T_+$ . In other words, the tipping status is, in this case, determined by  $T_{+}$  only. When the temperature varies in time, as for our emissions scenarios, tipping can be avoided even if the temperature exceeds the critical threshold  $T_+$ , provided that the overshoot time is short compared to the intrinsic timescale of the element (Ritchie et al., 2019, 2021).

This is illustrated in Figure 4, which depicts the evolution of AMOC and GRIS under the 2000 PgC emission scenario, with both elements having the same critical temperature threshold,  $T_+ = 2$ °C. Initially, the temperature rises quickly due to anthropogenic emissions and peaks at 4.2°C, well above the critical threshold for AMOC and GRIS, then gradually declines due to natural CO<sub>2</sub> sinks. AMOC tips, as the overshoot duration is long relative to its internal timescale, whereas GRIS remains untipped due to its slower dynamics. In such cases, the tipping status of an element depends both on  $T_+$  and  $\tau_-$ .

In overshoot scenarios, Ritchie et al. (2019) showed that an element will avoid tipping if

$$4a_0^2\kappa(q_{\text{max}} - q_+)t_{\text{over}}^2 

Figure 4. (a) Temperature evolution ( $\delta T_U$ ) for the scenario with 2000 PgC of cumulative CO<sub>2</sub> emissions added after 2024. The x-axis has a logarithmic scale. (b, c) Evolution of two tipping elements, GRIS and AMOC, under the same scenario, with the same critical thresholds but different intrinsic timescales. Both elements overshoot their critical threshold, but only AMOC tips, because it is a fast tipping element. These examples are from the decoupled experiment (i.e., no interactions between elements and no feedbacks from elements on temperature).

theory predicts that tipping elements with such parameter combinations will tip under that scenario. Conversely, parameter pairs above the threshold indicate that tipping elements will not tip under the associated emission scenario.

As an example, we plotted the parameter combinations of GRIS and AMOC from Figure 4. In this case, the pair  $(\tau_{-,GRIS}, T_{+,GRIS})$  lies above the theoretical threshold for the 2000 PgC emission scenario, and GRIS does not tip under this scenario. Conversely, the pair  $(\tau_{-,AMOC}, T_{+,AMOC})$  lies below the same threshold, and AMOC tips under that scenario. In this instance, the theoretical prediction aligns with the observed tipping outcomes.

To assess whether the Ritchie theory holds more generally, we compare the theoretical thresholds with the empirical tipping thresholds (solid lines in Figure 5a), which were determined using a bisection algorithm. Overall, for the scenarios considered, the theoretical thresholds align well with the observed tipping thresholds. This alignment is particularly strong for tipping elements with timescales shorter than or comparable to the timescale of global warming (~100 years), where the assumptions made to derive inequalities 35 and 38 hold true (Ritchie et al., 2019).

Based on the tipping thresholds, we observe that AMOC and AMAZ are likely to tip if peak warming exceeds their critical temperature threshold. As they are fast-tipping elements (with low  $\tau_{-}$ ), even a small and brief overshoot leads to tipping.

Figure 5. (a) Tipping thresholds for the tipping elements, as a function of their critical temperature  $T_+$  and their intrinsic time scale  $\tau_-$ . The gray rectangles indicate the range of possible parameter values for each element. The plain coloured lines represent the tipping thresholds for different scenarios in the decoupled experiment (interactions between tipping elements and their feedback on temperature are not included). An element with a  $(T_+, \tau_-)$  combination that is below a given threshold will tip for this scenario. The coloured dotted lines are the theoretical predictions for the tipping thresholds based on the criterion from Ritchie et al. (2019), see Eq. 38. (b, c) Zoom on the tipping threshold for the scenario with 2000 PgC of cumulative emissions, and for the AMOC and GRIS. Each dot represents a combination of parameters  $(T_+, \tau_-)$  from our ensemble, and its colour identifies the element tipping status. Panel (b) is for the decoupled experiment, while panel (c) is for the coupled experiment. In the latter case, the tipping threshold, which was computed for the decoupled case, doesn't neatly separate tipping and non-tipping behaviours. This indicates that including interactions between elements and their feedbacks on temperatures modifies the effective tipping threshold.

This contrasts with slower elements. For example, WAIS and GRIS are unlikely to tip under the 1000 PgC scenario, where temperatures peak at  $2.9^{\circ}$ C. Even in the 4000 PgC scenario, where peak warming reaches  $6.3^{\circ}$ C, well above the highest possible threshold for GRIS and WAIS ( $T_{+}=3^{\circ}$ C), they may still avoid tipping. For GRIS, this stability for high warming scenarios has been confirmed in state-of-the-art ice sheet models, provided that temperatures decrease sufficiently fast after peaking (Bochow et al., 2023; Höning et al., 2024).

In the decoupled experiment, the computed tipping threshold effectively separates between tipped and non-tipped elements, by construction (Figures 5b). In this case, the tipping risk for an element under a given scenario—defined as the percentage


**Figure 6.** Tipping risk for the different tipping elements, as a function of cumulative emissions and for different experiments (baseline, coupled, interactions-only, feedbacks-only). For a given element and scenario, the tipping risk is defined as the percentage of ensemble simulations generated with different parameter sets in which this element tips. For reference, the peak warming corresponding to a given level of emissions in the decoupled experiment is indicated on the upper axis of the plots.

of sampled parameter combinations that lead to tipping—approximates the fraction of the element's possible parameter space (represented by the gray rectangle) that falls below the scenario's tipping threshold.

In the coupled experiment, interactions between elements and their feedbacks on temperature cause the AMOC and GRIS to tip in cases where they would otherwise not tip (Figure 5c). For GRIS, we also observe the opposite effect, with cases where it tips in the decoupled experiment but doesn't in the coupled one, indicating that interactions may both destabilise and stabilise elements. Identifying a clear tipping threshold in the  $(T_+, \tau_-)$  parameter space is no longer possible as the tipping behaviour now depends on additional parameters such as the coupling coefficients  $\epsilon_{ij}$ . Nevertheless, we can still compute the tipping risk.

Figure 6 shows the tipping risk of each element as a function of cumulative emissions across four experiment setups: decoupled, interactions-only, feedbacks-only, and coupled. By comparing the coupled experiment to the feedback-only, and the interactions-only experiment to the decoupled experiment, we observe that interactions consistently increase the tipping risk for WAIS, EASB, and AMOC across all emission scenarios, with the most pronounced effects at intermediate emission levels (2000-3000 PgC). In the lower emission scenarios (<1000 PgC), tipping events are rare, so interactions have minimal impact. Under the higher emission scenarios (>4000 PgC), elements have a high probability of tipping regardless of whether





we consider interactions. For GRIS, interactions reduce the tipping, due to the strong stabilising influence of AMOC. For the AMAZ, the risk differs little whether interactions are considered or not. AMOC is the only element affecting the AMAZ, and its impact can be stabilising or destabilising, meaning in this case that interactions have a minimal effect on aggregated tipping risk, even though they can significantly affect individual simulations. In emission scenarios with more than 3000 PgC of cumulative emissions, interactions diminish the AMAZ tipping risk slightly. Indeed, AMAZ tipping becomes nearly inevitable without AMOC influence, which then only impacts the tipping outcome when it is has a stabilising effect, reducing the aggregated tipping risk. The EAIS only tips once in the coupled experiment, for the highest emission scenario (5000 PgC) and for a unique parameter set from our ensemble. This stability reflects its combination of a high temperature threshold and slow dynamics (see Figure 5), as well as the fact that in this model it is not directly affected by other elements, except indirectly through global mean temperature.

Comparing the coupled experiment to the interactions-only, and the feedbacks-only experiment to the baseline, shows that including feedbacks of elements on global mean temperature always increases the tipping risk. Indeed, all else being equal, increasing temperatures has a destabilising impact on elements in our model ( $c_1 < 0$  by design, see equations 3,7, and 13). While the impact of feedbacks on tipping risk is smaller than that of direct interactions, it can still be substantial for a given scenario and element, increasing tipping risk by up to 10 percentage points. In appendix D, we further test tipping risk sensitivity to an increased ECS, doubled interaction strength, and the exclusion of monostable elements ASI and PERM.

In figure 7, we present the tipping risk for each element in both the decoupled and coupled experiments using "burning embers" diagrams, and compare with the tipping risks that would result from constant temperature scenarios. Under these constant temperature scenarios, the risk only depends on the warming and the critical temperature threshold  $T_+$  of the elements, which is often the assumption made, explicitly or implicitly, when such "burning embers" plots appear in the literature and media (see for example: Schellnhuber et al., 2016; Wunderling et al., 2021; Armstrong McKay et al., 2022). This communicates risk simply and efficiently, but, as we show it, misrepresents the risk for the slower elements. Indeed, the tipping risk in the decoupled experiment for GRIS, WAIS, EASB, and EAIS is significantly lower than that predicted by a constant temperature scenario with equivalent peak warming. In the coupled experiment, the risk further changes as described above, highlighting the importance of considering both non-stationary temperature dynamics and interactions between the elements.

## 3.2 Tipping cascades

In this section, we compare the tipping outcomes in all pairs of successive scenarios, i.e., scenarios that are separated by only 50 PgC of CO<sub>2</sub> emissions. This allows us to define *tipping cascades*. We order scenarios by increasing emissions and index them such that scenario j corresponds to the scenario where  $j \cdot 50$  PgC of CO<sub>2</sub> emissions are added after 2024. In a given experiment and for a given set of parameters, we consider that there is a *tipping event* for scenario j if at least one tipping element has changed status in scenario j compared to in scenario j - 1. If only one element has changed status, we classify the event as a *single tipping event*. If more than one element has changed status, we call it a *tipping cascade*. The number of elements whose status has changed is the *size of the cascade*. If all the elements whose status has changed went from "not tipped" for scenario j - 1 to "tipped" for scenario j, we call it a *destabilising cascade*. If at least one element of the cascade

Figure 7. Tipping risk as a function of peak warming for the decoupled and coupled experiments, compared to the tipping risk that would result from constant temperature scenarios. In the latter case, the constant temperature is also the peak temperature, and the tipping risk is: 0% if the temperature T is lower than the minimum possible temperature threshold for the element  $T_{+,i}^{min}$ ; 100% if the temperature T is above the maximum possible temperature threshold  $T_{+,i}^{max}$ ; and equal to  $100 \cdot (T - T_{+,i}^{min})/(T_{+,i}^{max} - T_{+,i}^{min})$  if  $T_{+,i}^{min} \leq T \leq T_{+,i}^{max}$ . In the decoupled and coupled experiments, the tipping elements experience a time-varying temperature, and the relationship between peak warming and tipping risk is more complex. In the coupled experiment, peak warming furthermore depends on the state of the elements and may not always be equal for the same emission scenario. To simplify the analysis, we have plotted for both experiments the risk as a function of cumulative emissions, and indicated on the y-axis the corresponding peak warming from the decoupled experiment, where warming does not depend on the state of the tipping elements. Note that the difference in peak warming between the decoupled and coupled setups for a given model run is rather small and stays below  $0.5^{\circ}$ C (see Appendix F).

went from "tipped" to "not tipped", we call it a *stabilizing cascade*, regardless of the number of elements that were destabilised. Figure 8 shows an example of a size-3 stabilising cascade occurring for the 2150 PgC scenario in the coupled experiment.

Causality is not built into our definition of a tipping cascade: two elements that change tipping status under the same scenario j are considered part of the same tipping cascade even if one does not *cause* the tipping of the other, either directly or indirectly (e.g., via the temperature or other elements). Establishing the precise causal structure of cascades would require a substantial number of targeted experiments that selectively enable and disable specific interactions and feedbacks, a level of analysis out of the scope of this study.

#### 3.2.1 Which cascades do we observe?



A total of 4114 tipping events occur in the coupled experiment. These events are shown in Figure 9, sorted by their frequency of occurrence. Single tipping events are the most common, accounting for 78% of all events. The remaining 22% of events are tipping cascades of sizes 2,3,4, or 5. We observe no cascades of size 6. The most frequent cascades involve either the AMOC and the ice sheets or only the ice sheets. Stabilizing cascades make up 23.8% of all cascades, primarily driven by the stabilization of the GRIS due to the AMOC's collapse. Notably, we only observe 1 cascade where the AMAZ is stabilised by the AMOC, despite a stabilising AMOC $\rightarrow$ AMAZ interaction being as likely as a destabilising one in our model. This is because

**Figure 8.** Example of a tipping cascade happening in the coupled experiment for a given set of parameters. In the 2100 PgC scenario, GRIS, WAIS, and AMAZ are tipped. When 50 PgC of emissions are added, AMOC and EASB additionally tip, while GRIS does not tip anymore due to the influence of the AMOC. In this case, we have a size-3 stabilising cascade for the 2150 PgC emission scenario. In the 2150 PgG scenario, AMAZ returns to its baseline state after 50 kyrs because of the stabilisation effect from AMOC, and the temperatures slowly decreasing due to the natural uptake of CO<sub>2</sub> by carbon sinks. Nevertheless, AMAZ is counted as "has tipped" since it reached its collapsed state for some time.

the AMAZ is a fast tipping element, and in a scenario where the AMOC changes tipping status, the latter generally tips too late to prevent the AMAZ from tipping. For similar reasons, we do not observe cascades where GRIS tipping leads to the AMOC tipping, despite it being often mentioned as a typical example of tipping cascade (Steffen et al., 2018; Lenton et al., 2019). This absence also results from our definition of a tipping cascade, which requires changes between two very close scenarios. If the GRIS doesn't tip in scenario j-1, but tips in scenario j due to the slight increase in temperature, its trajectory will pass very close to the bifurcation point, where the dynamics are slow. This has two consequences: (1) the meltwater flux produced ( $\propto x_{GRIS}$ ) is small, with a limited impact on AMOC; (2) if the AMOC does tip, it will do so on a shorter time scale than GRIS, and its stabilising influence will be enough to revert GRIS's tipping, excluding GRIS from the cascade count. However, not observing a destabilising GRIS-AMOC cascade does not mean that GRIS has no influence on AMOC. As shown in Figure 6, interactions with other elements, including GRIS, increase the tipping risk for AMOC. Overall, these specific examples emphasise once more the critical role of time scales when considering realistic scenarios with non-stationary temperatures.

#### 490 3.2.2 At which levels of emissions do we observe most cascades?

We observe that the majority (92.5%) of cascades happen for scenarios with between 350 PgC and 3000 PgC of added cumulative emissions, or with between  $\sim$ 2°C and  $\sim$ 5.4°C of peak warming (see Figure 10a). About 2.3% of observed cascades occur in scenarios with emissions up to 300 PgC. In the decoupled experiment, these scenarios lead to peak warming that stays below 2°C, meaning that there is a possibility for tipping cascades even if the Paris Agreement target is met. We observe very few cascades happening for high emissions scenarios, as it is very probable that all tipping elements have already tipped by then.

Figure 10b shows the risk of various tipping events occurring for scenarios with cumulative emissions lower than or equal to a given value, in the coupled experiment. For a given event type and emissions level X, the risk is here defined as the


Figure 9. Tipping events frequency for the coupled experiment. The dots represent the tipping status change of each element. For a tipping event happening in scenario j, a red dot indicates that the element does not tip in scenario j-1 but tips in scenario j (destabilisation); a blue dot indicates that the element tips in scenario j-1 but does not tip in scenario j (stabilisation); and a grey dot indicates that the element doesn't change tipping status between two successive scenarios. The inset table shows the fraction of events that are classified as single tipping events, cascades, stabilising cascades, or destabilising cascades. When two numbers are given, the first one indicates the percentage over all tipping events, and the second one indicates the percentage over all tipping cascades.

percentage of parameter sets that lead to at least one such tipping event in any scenario with cumulative emissions less than or equal to X. Figure 10b is similar but not equivalent to the "integration" of Figure 10a, as for a given parameter set, the same tipping events may occur in multiple scenarios. We find that the risk of observing tipping events occurring in scenarios with cumulative emissions of up to 300 PgC-scenarios that meet the Paris agreement- is 6.7%, mainly due to single tipping events. The risk of observing a size 2 or size 3 cascade is 1.9% and 0.2%, respectively.

The risk of tipping events occurring increases rapidly with emissions: for scenarios up to 1200 PgC, the overall risk reaches 69.9%, while the risk of experiencing a size-2 or size-3 destabilising cascade increases to 19.1% and 4.2%, respectively. For these scenarios, global mean temperature in 2100 CE remains at or below 2.7 °C above pre-industrial levels, the warming expected under current policies (Climate Action Tracker, 2024). The size-2 cascades that dominate under such conditions most often involve AMOC and WAIS, AMOC and EASB, or GRIS and WAIS, while size-3 cascades typically involve AMOC, WAIS, and EASB. These combinations are also the most frequent destabilising cascades when considering all emission scenarios (see Figure 9). By contrast, the risk of a stabilizing cascade in scenarios with cumulative emissions below or equal to 1200 PgC remains low at 2.4% (see also Figure 10a). This is because such cascades generally involve ice-sheet stabilisation by the AMOC, which requires a starting situation with a tipped ice sheet and an untipped AMOC, a condition that rarely


Figure 10. Timing of tipping events. A tipping event is considered to happen for scenario j if some tipping elements change status compared to scenario j-1. (a) Distribution of tipping cascades as a function of cumulative emissions in the coupled experiment. (b) Risk of observing a given tipping event before a given amount of cumulative emissions. For reference, the peak warming corresponding to a given level of emissions in the decoupled experiment is indicated on the upper axis of the plots. The shaded areas correspond to scenarios with cumulative emissions after 2024 lower than or equal to 300 PgC (Paris agreement) and 1200 PgC (current policies).

occurs in lower emissions scenarios. In appendix D, we examine how these tipping risks change for higher equilibrium climate sensitivity, stronger interactions, and with the removal of ASI and PERM.

## 3.2.3 What are the roles of elements in a cascade?

Understanding the roles of tipping elements within a cascade, especially those that initiate it, is instructive. It is valuable from a theoretical perspective to clarify the dynamics of cascades, but also from a very practical standpoint, as these elements should be carefully monitored in priority, and if possible managed.

Wunderling et al. (2021), define the initiator of a tipping cascade as the element of the cascade with the critical temperature threshold closest to the constant temperature forcing for which the cascade occurred. With this definition, they find that GRIS and WAIS are often the initiators of cascades, which results from them having the lowest critical temperatures. Using the same model as Wunderling et al. (2021), but different analysis techniques, Rosser et al. (2024) reach the similar conclusion that GRIS and WAIS are the most decisive elements influencing tipping risks and cascades. However, both studies use constant temperature forcings, under-representing the influence of elements' timescales, as shown in section 3.1. To address this, we take a different approach, outlined below.

We define an element i as a *potential initiator* of a tipping cascade occurring under scenario j if it changes tipping status solely due to the increase in anthropogenic emissions from scenario j-1 to j. This can be verified by rerunning the model under scenario j with all other elements  $k \neq i$  held at their values from scenario j-1. In other words, a potential initiator tips "on its own" and not because of changes in other elements from scenario j-1 to scenario j. However, it may not necessarily be




**Figure 11.** Occurrence and roles of the tipping elements in cascades for the coupled experiment. The left plot is for all cascades. The middle and right plots consider the specific cases of destabilising and stabilising cascades.

the primary driver of a cascade, hence the denomination of *potential* initiator. Indeed, the cascade could also be driven by other elements that change tipping status under the same scenario, or even by elements not counted as part of the cascade, including sea ice and permafrost, that have been sufficiently altered by the emissions increase to influence the tipping status of others. In practice, this means that a cascade can have multiple potential initiators, or even none. Again, we choose to work with this definition despite its limitations because establishing the precise causal structure of a cascade is challenging.

We observe that AMOC is the element that appears the most in cascades, followed by GRIS, EASB, and WAIS. This is not surprising as these elements are tightly coupled (see Figure 3a). We find that AMOC is a potential initiator in most of the cascades it appears in, and in about 50% of all cascades, which makes it the most frequent potential initiator overall (see Fig. 11). The next most frequent potential initiator of cascades is GRIS, which is a potential initiator in about 18% of all cascades, and in a bit more than a third of the cascades it appears in. AMAZ appears in fewer than 16% of cascades but serves as a potential initiator in 44% of them, despite only affecting other elements through a relatively small feedback on global mean temperature. A deeper analysis shows that in about 40% of the cascades where AMAZ is a potential initiator, another element also acts as a potential initiator, illustrating the limitations of our approach in determining the precise causal sequence of events within a cascade.

We obtain some further insights by separating the analysis between stabilising and destabilising cascades. AMOC and GRIS appear together in all but one stabilising cascade. Since GRIS is stabilised, it cannot, by definition, be a potential initiator in these cascades. In contrast, it is a potential initiator of most destabilising cascades it appears in. On the other hand, AMOC is a frequent potential initiator in both stabilising and destabilising cascades, underlining its significant role in tipping dynamics.

## 4 Discussion

Our objective was to explore tipping cascades using a simple modelling framework that integrates essential physics and the latest knowledge on tipping elements and their interactions. The extended version of SURFER presented here reflects that






approach, providing a fast model with a dynamic carbon cycle, enabling us to conduct multi-millennial, emission-based simulations. The parametrization of nonlinear elements is flexible, accommodating 6 bistable tipping elements (GRIS, WAIS, EASB, EAIS, AMOC, and AMAZ), as well as 2 mono-stable nonlinear elements (ASI and PERM). This is, in total, twice as many elements as existing similar studies (Wunderling et al., 2021, 2023). Key parameters—such as critical temperature thresholds, timescales, interaction strengths, and feedback effects on global mean temperature—are calibrated based on the most recent literature. Uncertainties in these parameters are taken into account using a large Monte-Carlo ensemble.

With this framework, we explored tipping dynamics and cascades across a range of emission scenarios. Our findings highlight that, in addition to critical temperature thresholds, the intrinsic timescales of individual elements play a substantial role in determining tipping risk. Slow-tipping elements, in particular, can avoid tipping even under substantial and prolonged overshoots of their critical temperature thresholds. While these findings align with well-established theory (Ritchie et al., 2019) and observed behaviour in both conceptual models (Wunderling et al., 2023) and complex model studies (Bochow et al., 2023; Höning et al., 2024), we note that this aspect is often overlooked in risk assessments, which tend to focus mainly on temperature thresholds and the likelihood of crossing them under projected warming.

Our results show that including interactions between elements increases the tipping risk for WAIS, EASB, and AMOC, while reducing the tipping risk for GRIS, in line with Wunderling et al. (2021). Including feedbacks from tipping elements on global mean temperatures raises the tipping risk across all elements, though this effect remains secondary to the influence of interactions. This is consistent with (Deutloff et al., 2025), who find only a limited role for such feedbacks in driving tipping. Overall, we found a 69.9% risk of observing at least one tipping event under scenarios where warming in 2100 is lower than or equal to 2.7°C, the level expected under current policies. This result is comparable to earlier work that considered interactions only (Wunderling et al., 2023; Möller et al., 2024) and mainly reflects the risk of single tipping events. Cascades can still occur under these scenarios, but at lower frequencies than reported by Wunderling et al. (2021), who found a majority of cascades occurring within a 1–3 °C warming range. These differences arise from our use of realistic, non-stationary scenarios rather than constant-temperature scenarios. Importantly, our results suggest that meeting the Paris Agreement target of limiting warming to below 2 °C would reduce the risk of observing tipping events and cascades by roughly an order of magnitude compared to the risks under current policies.

We found AMOC to be the main potential imitator of cascades. This is in contrast with the results from Wunderling et al. (2021), who identified GRIS and WAIS as the main initiators of cascades. One of the reasons for this difference is that we do not observe cascades where GRIS destabilises the AMOC. These cascades, frequent in Wunderling et al. (2021), would provide numerous instances where GRIS acts as an initiator while AMOC does not. It is probable that Wunderling et al. (2021) observes a lot of these cascades because they model the impact of GRIS on AMOC as proportional to  $x_{GRIS}$ , whereas we model it as proportional to  $\dot{x}_{GRIS}$ .

While our definition of a tipping cascade is similar to the one used by Wunderling et al. (2021), it differs slightly from the term's broader use in the literature. By comparing tipping status across closely spaced emission scenarios, our approach highlights how even minimal changes in forcing can result in distinct Earth system states, where one, two, three, or more elements have tipped. We use scenarios separated by 50 PgC of emissions, which corresponds to only five years of emissions at





today's rates. This type of cascade aligns with definitions of joint or domino cascades by Klose et al. (2021). However, if only one element changes status for a given scenario in the coupled experiment but would not have tipped under the same scenario in the decoupled experiment, we do not count it as a cascade, despite the element tipping with the aid of interactions. This would align with the definition of a two-phase cascade by Klose et al. (2021), where the tipping of one element brings another one closer to its bifurcation point, but additional forcing is needed to trigger its tipping.

While the results of this study provide insights into tipping dynamics and cascades, they inevitably rest upon certain assump-

tions and simplifications. Using a unique equation to represent all elements allows a streamlined analysis and easy comparison between elements, which differ by only a few parameters. However, this equation does not fully capture the complex dynamics of real-world tipping elements. For example, it only allows for bifurcation-tipping, i.e., the forcings needs to go past he bifurcation point to trigger tipping. For some elements like the AMOC, the rate of forcing change may also have an effect on whether the element tips or not, which may impact the cascading dynamics with the ice sheets (Sinet et al., 2023; Klose et al., 2024). Tipping elements may also have a multitude of intermediate states (Bastiaansen et al., 2022). This was shown in some models for the GRIS (Höning et al., 2023) and the AMOC (Lohmann et al., 2023), or in vegetation systems (Rietkerk et al., 2021). Additionally, real-world interactions between elements are certainly more complicated than simple linear dependencies. The parameter calibration could also be improved. Due to the lack of information available, we have fixed arbitrarily the values of  $T_-$ ,  $T_-$ , and  $T_+$ . A better calibration of the "back tipping" points ( $T_-$ ,  $T_-$ ) would enable the study of the long term reversibility of cascades. For these matters, we encourage the community to report more on these quantities, for example in future literature reviews and assessments. Furthermore, the calibration of the coupling coefficients based on the assessments of interaction strengths from Wunderling et al. (2024) is somewhat arbitrary. As explained in section 2.4.2, targeted experiments in more complex models with isolated forcings could help to provide better calibration. Future experiments made in the contexts of projects such as TipESM or TIPMIP may also provide further constraints on the distributions of the critical thresholds  $T_+$ 

Finally, it should be noted that the impacts of tipping elements may take several centuries to unfold, or even millennia. In this study, we only looked at whether the elements tipped or not during the  $\sim$ 100 000 year span of the model runs, but didn't focus specifically on the tipping times or impacts. When we talk about cascade that occur for scenarios with 2100 warming below 2.7°C, it doesn't mean that all the elements will have transitioned to their collapsed state by the time the temperature reaches 2.7°C. We'll examine the impact of cascades in a follow-up study, specifically their influence on global mean temperatures and sea level rise, to examine if they could lock us on a pathway towards a hothouse Earth.

and timescales  $\tau_-$ . Using formal distributions for the parameters would allow to interpret our tipping risks as probabilistic risk

## 5 Conclusions

assessments.

In this study, we integrated the latest understanding of tipping elements and their interactions within an adaptable modelling framework to explore tipping cascades.





Our results demonstrate the important role of intrinsic time scales and the need for realistic, non-stationary scenarios in assessing tipping risk. Notably, slow tipping elements such as the ice sheets may remain stable even under relatively large and prolonged overshoots of their critical temperature thresholds, suggesting that assessments focused solely on critical temperature exceedances may overestimate their tipping risk.

On the other hand, we found that interactions among elements generally increase tipping risks, while feedbacks from tipping elements on global mean temperature lead to a further, but comparatively smaller increase. We found that most cascades happen for scenarios with peak warming between 2.0°C and 5.4°C. While tipping events and cascades remain possible below 2 °C, the modelled risk is an order of magnitude lower than under current-policy pathways, underscoring the need for urgent action to meet the Paris Agreement targets. In contrast to previous studies, we identified the AMOC as the primary potential initiator of cascades, which is concerning given that some studies have shown it could tip within this century even under intermediate warming scenarios.

While our model offers valuable insights into potential tipping phenomena under various conditions, the simplifications made inherently limit its ability to fully capture the complexity of real-world tipping dynamics. Consequently, these results should be viewed as illustrative rather than predictive. Nevertheless, we believe this extended SURFER version has its place within the modelling hierarchy: it not only generates hypotheses that can be explored in more complex models but also stands to benefit from ongoing and future research efforts, such as TIPMIP and TipESM. These initiatives will enhance our understanding of tipping dynamics and provide better constrained input data for SURFER, further supporting its development as an emulator and as a tool for probabilistic risk assessments of tipping events.

Code and data availability. The version of SURFER used to produce the results showed in this paper is archived on Zenodo (https://zenodo.org/records/17279674, Couplet, 2025), as is the input data to run the model and most of the data to produce the plots. The code of SURFER is licensed under MIT license. Historical CO<sub>2</sub> emissions are from Friedlingstein et al. (2025) and are available at https://globalcarbonbudgetdata.org/latest-data.html, last access: 10 September 2025. Historical CH<sub>4</sub> emissions are from Jones et al. (2023) and are available at https://zenodo.org/records/10839859, last access: 10 September 2025. CO<sub>2</sub> and CH<sub>4</sub> emissions for the SSP scenarios are available in the SSP database hosted by the IIASA Energy Program at https://tntcat.iiasa.ac.at/SspDb, last access: 10 September 2025.

## Appendix A: Emission scenarios

We use  $CO_2$  emissions data from the Global Carbon Budget (Friedlingstein et al., 2025) for the period 1850 to 2024, incorporating fossil fuel emissions, land-use change emissions, and the cement carbonation sink. For simplicity, we treat historical land-use emissions as fossil emissions, although SURFER allows for distinctions between emission types. Cumulative  $CO_2$  emissions from 1850 to 2024, including land-use emissions, amount to 723 PgC. From 2024 to 3000 CE, emissions E(t) follow the logistic equation

$$E(t) = -\frac{dF(t)}{dt} = -aF(t)(F(1850) - F(t))$$
(A1)

Figure A1. Comparison between  $CO_2$  emissions from SSP scenarios (land-use + fossil) with scenarios used for this paper. In the experiments we perform, we force the model with 10 emissions scenarios, starting with 50 PgC of total cumulative emissions after 2024 CE, adding 50 PgC of  $CO_2$  emissions for each successive scenario until we reach 500 PgC of cumulative emissions. In this figure, only one in ten scenarios is plotted.

similarly as in Winkelmann et al. (2015) or Lord et al. (2016). Emissions before 1850 and after 3000 CE are set to zero. Here, F(t) represents the total emissions remaining to be released at time t, and a is a parameter defined by

$$a = \frac{-E(2024)}{F(1850)(F(1850) - F(2024))}$$
, (A2)

where E(2024) are the historical CO<sub>2</sub> emissions in 2024 and F(2024) is the amount of total emissions still to be released after 2024.

In this study, we explore a range of 100 emissions scenarios, with F(2024) varying between 50 PgC and 5000 PgC. Indexing each scenario by j, we define  $F_j(2024) = j \cdot 50$  PgC, for  $j \in 1, 2, \dots, 100$ . Solving equation A1 for F, we find

$$F(t) = \frac{F(t^*)F(1850)e^{aF(1850)(t-2024)}}{F(2024)e^{aF(1850)(t-2024)} + F(1850) - F(2024)}$$
(A3)

where the total cumulative emissions are given by F(1850) = 723 PgC + F(2024). Throughout this paper, we often refer to scenarios by their cumulative emissions value after 2024, meaning that when we mention a scenario with cumulative emissions X, we refer to the scenario j with  $F_j(2024) = X \text{ PgC}$ . Our scenarios cover a similar range to that of the SSP scenarios used in IPCC reports, ensuring their policy relevance (see Figure A1).

We use the same CH<sub>4</sub> emissions for all scenarios. For emissions from 1850 up to 1989 (included), we use the data from Jones et al. (2023) and include fossil and land-use emissions. From 1990 to 2100, use the values for the SSP1-2.6 scenario as provided in the SSP database (https://tntcat.iiasa.ac.at/SspDb/dsdRiahi et al. (2017); Gidden et al. (2019)). From 2100 to 2300, we use the values for the SSP1-2.6 scenario extension, as described in Meinshausen et al. (2020). Outside of this range, CH<sub>4</sub> emissions are set to zero. This approach results in a total cumulative CH<sub>4</sub> emission of 44 PgC.







#### Appendix B: Monte-Carlo ensemble and latin hypercube sampling

There is substantial uncertainty surrounding various aspects of tipping elements and monostable elements—for instance, their critical temperatures and intrinsic time scales. Arguably, even greater uncertainty exists regarding the strength of their interactions. To account for this, we run our model across a wide range of parameter sets. Since we evaluate each parameter configuration under 100 different emission scenarios, the total computation time increases rapidly, and we need to explore the parameter space efficiently. To do that, we employ Latin Hypercube Sampling (LHS).

We sample the values for  $T_+$  or  $T_I$ ,  $\tau_-$ , and for the coupling coefficients  $\epsilon_{ij}$  in uniform distributions whose bounds are given in Table 1. We want an ensemble of  $N_{\rm params}$  parameter sets. With Latin Hypercube Sampling, each parameter range is divided into  $N_{\rm params}$  equal parts, and a parameter value is sampled in each interval exactly once. This ensures a more representative sampling across the entire high-dimensional parameter space ( $n_{\rm dim}=31$ ) than simple random sampling. We perform the Latin Hypercube sampling using the LatinHypercube class from the Quasi-Monte Carlo submodule of SciPy, with the random-cd optimisation scheme.

To determine the number of parameter sets  $N_{\rm params}$  necessary to adequately span the parameter space, we test the variability of model results for experiments using parameter ensembles of varying sizes, from  $N_{\rm params} = 10$  to  $N_{\rm params} = 1000$ . For each value  $N_{\rm params}$ , we create 10 independent ensembles of  $N_{\rm params}$  parameter sets using Latin Hypercube sampling, and for each of these ensembles, we compute the tipping risk of the individual tipping elements under 6 emissions scenarios. As a reminder, the tipping risk for a given parameter ensemble, scenario, and tipping element is defined as the fraction of runs where this element tips, across all parameter sets in the ensemble.

We find that the mean tipping risk from 10 experiments with ensembles of  $N_{\rm params}$  parameter sets converges as  $N_{\rm params}$  grows and that the variability decreases (see Figure B1). For  $N_{\rm params}=100$ , the results already show limited variability, with tipping risk for the selected scenarios varying by no more than 12 percentage points across the different experiments. For  $N_{\rm params}=1000$ , the tipping risk varies by no more than 3.1 percentage points between experiments. This demonstrates that the experiments with  $N_{\rm params}=1000$  parameter sets described in the main text provide robust results for the tipping risk of individual elements, in the sense that the quantitative results would not change substantially if someone repeated the experiments with a different ensemble of 1000 parameter sets.

We did not test the variability of the results concerning cascades for different parameter ensembles. This would require running test experiments using the full set of 100 emissions scenarios, which is computationally costly. However, with 904 cascades occurring in the coupled experiment described in the main text for  $N_{\text{params}} = 1000$ , we are confident that our results for cascades are also robust, at least qualitatively. We expect that experiments using different ensembles of  $N_{\text{params}} = 1000$  parameter sets would yield a similar set of tipping events as the ones described in Figures 9, 10, and 11, and in Table D1.

# **Appendix C: Transition time scales**

We sample the timescales  $\tau_{-,i}$  from uniform distributions with bounds  $\left[\tau_{-,i}^{\min},\tau_{-,i}^{\max}\right]$  selected to ensure that the effective transition timescales of the tipping elements fall within or close to the assessed range reported in the literature.

Figure B1. Variability of tipping risk as a function of parameter ensemble size ( $N_{\text{params}}$ ). Tipping risk is computed from experiments using parameter ensembles of varying sizes, in a coupled setup (i.e., including interactions between nonlinear elements and their feedbacks on temperature). Ensemble sizes used are  $N_{\text{params}} = 10, 14, 19, 27, 37, 52, 72, 100, 139, 193, 268, 373, 518, 720, 1000$ . For each value of  $N_{\text{params}}$ , ten experiments are conducted using independently sampled parameter ensembles. In each experiment, tipping risk is evaluated for six emission scenarios (50, 1000, 2000, 3000, 4000, and 5000 PgC of cumulative  $CO_2$  emissions after 2024). Solid lines indicate the mean tipping risk across the ten experiments, while shaded areas represent the full range of results.

Here is how we determine the bounds  $\left[\tau_{-,i}^{\min},\tau_{-,i}^{\max}\right]$ . First, we define, in our model, the *transition time scale* of a tipping element i as the time it takes to go from  $x_i=x_+$  to  $x_i=x_-$  when tipping. We then proceed with calibration experiments where we vary the internal time scales  $\tau_{-,i}$  of elements while keeping their critical temperature thresholds  $T_{+,i}$  fixed to the best estimates from Armstrong McKay et al. (2022) (listed in Table 2).

To fix  $\tau_{-,i}^{\min}$ , we simulate element i under a constant temperature forcing  $T = T_{+,i} + 3^{\circ} \text{C}$  with no couplings (decoupled setup). We then set  $\tau_{-,i}^{\min}$  as the value of  $\tau_{-,i}$  that produces a simulated transition time scale equal to the lower bound reported in Table 2. To fix  $\tau_{-,i}^{\max}$ , we proceed similarly, except that we use a constant temperature forcing  $T = T_{+,i} + 1^{\circ} \text{C}$  and that we search for the value of  $\tau_{-,i}$  such that the computed transition time scale is equal to the upper bound provided in Table 2.

Figure C1. Distribution of effective transition time scales for three different scenarios (1000 PgC, 3000 PgC, and 5000 PgC) in the decoupled experiment. The effective transition timescale is defined for an element that tips as the time it takes to transition from  $x = x_+ = 0.75$  to  $x = x_- = 0.25$ . The EAIS never tips in the decoupled experiment, which is why no distributions are plotted for this element. Vertical lines indicate the minimum, maximum, and best estimate from Armstrong McKay et al. (2022), also given in table 2.

The higher temperature forcing of  $T = T_{+,i} + 3^{\circ}\text{C}$  causes element i to tip more rapidly, justifying its use in estimating  $\tau_{-,i}^{\min}$ , the fastest (lowest) possible time scale. In contrast, the lower forcing of  $T = T_{+,i} + 1^{\circ}\text{C}$  leads to a slower tipping transition, hence its use in estimating  $\tau_{-,i}^{\max}$ , the slowest (highest) time scale.

For the nonlinear monostable elements (Arctic sea ice and permafrost), we define the transition time scale as the time it takes 710 to go from  $x_i = 1$  to  $x_i = x_I$ , and then follow the same calibration procedure for  $\tau_-$  as described above.

In the experiments presented in the main text, the effective transition time scale not only depends on  $\tau_{-}$ , but also on the choices  $T_{+}$ , the forcing scenario, and the couplings between the elements (for the coupled experiment). We find that for the majority of cases where elements tip, the transition timescale falls within the range assessed by Armstrong McKay et al. (2022), indicating that our calibration procedure is effective and effectively aligns with established estimates (Figure C1).

It is important to note that the fastest transition times in the decoupled experiment, approximately 300 years for GRIS, 200 years for WAIS, and 270 years for EASB, do not imply that these elements are fully collapsed by 2150 CE, 2050 CE, or 2120 CE. This is because our definition of transition time for an element i only covers the interval during which  $x_i$  evolves from  $x_+ = 0.75$  to  $x_- = 0.25$ . In the 5000 PgC scenario, which produces the fastest transitions, we find that complete collapse  $(x_i = 0)$  occurs no earlier than  $\sim$ 2730 CE for GRIS,  $\sim$ 2520 CE for WAIS, and  $\sim$ 2650 CE for EASB.




In our decoupled experiment with an equilibrium climate sensitivity of 3.5°C, the East Antarctic Ice Sheet (EAIS) does not completely melt, making it difficult to evaluate the effectiveness of our calibration procedure for this element. Additionally, the assessed range for its transition timescale is poorly constrained. Armstrong McKay et al. (2022) gives a minimum estimate of 10,000 years but do not provide a maximum estimate. For our calibration, we selected minimum and maximum transition timescales of 5,000 years and 15,000 years, respectively. This choice results in a sampling range for τ<sub>-,EAIS</sub> of [470,741] years, aimed at improving the simulation of the total sea level rise contribution from the Antarctic Ice Sheet. This situation also highlights that, since the effective transition timescale depends on T<sub>+</sub>, a lower sampling range for τ<sub>-</sub> does not necessarily yield shorter transition times when comparing different elements.

## Appendix D: Sensitivity analysis

In the main text, we have discussed the results of 4 experiments: the decoupled, interactions-only, feedbacks-only, and the coupled experiments. These experiments correspond to enabling or disabling the interactions between the elements and their feedback on temperature. In the following section, we test the sensitivity of our results to several other model settings: an increased equilibrium climate sensitivity, increased interactions strengths, and a removal of Arctic sea-ice and permafrost.

# D1 Equilibrium climate sensitivity (ECS)

IPCC assessed the very likely ECS range to be between  $2^{\circ}$ C and  $5^{\circ}$ C (IPCC, 2021). SURFER's ECS is determined by the climate feedback parameter (see Eq. 23), which we have chosen such that ECS =  $3.5^{\circ}$ C, a value in the middle of the assessed range. Here, we repeat the decoupled and coupled experiment but with ECS =  $5.0^{\circ}$ C, a value at the high end of the very likely range. To do this we set  $\beta = 0.78 \text{W m}^2 \,^{\circ}\text{C}^{-1}$ , while all other parameters are kept the same. Unsurprisingly, we observe that for a given scenario, the tipping risk is higher for a higher ECS (Figure D1). This is because a higher ECS leads to higher warming, and higher temperatures always increase the tipping risk in our model, all other things being equal. For AMOC and AMAZ, the tipping risk as a function of peak warming is almost identical for equivalent experiments with different ECS. This is because for these fast elements, their tipping status is mainly determined by short-term temperatures and whether warming exceeds their critical temperature thresholds (see section 3.1). For GRIS, WAIS, and EASB, the tipping risk as a function of peak warming is similar but slightly larger for the experiments with higher ECS (Figure D1). For those slow tipping elements, peak temperatures matter, but so does the rate at which temperature decreases after peaking (see Section 3.1). In scenarios with different climate sensitivities but the same peak warming, temperatures decrease more slowly on the millennial time scale when ECS =  $5^{\circ}$ C, as shown in Figure D2.

Overall, it shows that while the relationship between cumulative emissions and tipping risk may depend on ECS, the relationship between peak warming and tipping risk is more robust. Nevertheless, it should be noted that the relationship between peak warming and tipping risk can be impacted by anything that could change temperatures quite rapidly, such as methane emissions, strong negative CO<sub>2</sub> emissions or aerosol emissions.

**Figure D1.** Tipping risk of the different tipping elements, for the decoupled and coupled experiments, with ECS =  $3.5^{\circ}$ C or ECS =  $5^{\circ}$ C. For a given element and scenario, the tipping risk is defined as the percentage of ensemble simulations, generated with different parameter sets, in which the element tips. In the left panels, the risk is given as a function of cumulative emissions. In the right panels, the risk is given as a function of peak warming in the decoupled experiments, where warming does not depend on the state of the tipping elements.

## **D2** Interactions

Interactions between nonlinear elements are very poorly constrained, and admittedly, the rule we have chosen to assign values for the coupling coefficients as functions of the assessed interaction strengths and signs by Wunderling et al. (2024) is rather arbitrary (see Figure 3). With this rule, the value of the forcing induced by element i on element j for weak, intermediate and strong destabilising interactions, can reach up to 33%, 66%, and 100% the critical forcing value necessary to tip element j, respectively (or to push element j past its inflexion point in the case of monostable elements).

In Wunderling et al. (2021), who use a similar experimental setup as us and also have a parameter  $d_W$  that scales all interactions, the value of the forcing exerted by element i on element j for strong interactions can reach up to 5 times the



**Figure D2.** Temperature evolution in the decoupled experiment settings (interactions between tipping elements and their feedback on temperature are not included) for two different scenarios and ECS. Both scenarios result in the same peak warming and short-term evolution, but the multi-millennial evolution is different, which results in a different tipping risk for the slow tipping elements (see top 3 right panels of figure D1).

critical forcing value required to tip element j, and can trigger the tipping even for most of interactions which are considered as weak. A d=1 value in our model (see Eq. 16) approximately corresponds to a value  $d_W \sim 02$ . -0.4 in Wunderling et al. (2021). In Figure D3, we show the tipping risk for all elements in the coupled experiment but with d=2. In this case, the tipping of an element i is always sufficient on its own to trigger the tipping of element j in case of strong interactions, and may trigger tipping even in case of moderate strength interactions. This corresponds to a value of about  $d_W \sim 0.4 - 0.8$  in Wunderling et al. (2021). We observe that doubling all interaction strengths amplifies the effects of interactions already described in Section 3.1. Compared to the decoupled experiment, the AMOC, WAIS, and EASB tipping risk increases more for d=2 than for d=1, the GRIS tipping risk decreases more, and the AMAZ tipping risk doesn't change much, except for high emissions scenarios. The change in tipping risk resulting from interactions does not depend linearly on d, i.e., the difference in tipping risk for a given element and scenario between the decoupled and coupled experiments for d=2 is not twice the difference for d=1.

## D3 Monostable elements

We also examine tipping risk in the coupled experiment when Arctic sea ice (ASI) and permafrost (PERM) are excluded from the model. This exclusion leads to a significant reduction in the tipping risk for the AMOC, due to the destabilizing effects that both ASI and PERM have on the AMOC when included (Figure D3). This reduction in AMOC tipping risk subsequently propagates to other elements based on their dependence on AMOC: the tipping risk for WAIS and EASB decreases due to reduced destabilization from the AMOC, while the tipping risk for GRIS increases in its absence.

Interestingly, we observe that AMOC tipping risk is now only slightly higher than in the decoupled case, suggesting that in our model, the AMOC is only moderately impacted by GRIS and WAIS. It is expected that WAIS has a negligible net effect on AMOC tipping risk due to its potential for both stabilizing and destabilizing interactions. GRIS, on the other hand, exerts

Figure D3. Tipping risk of the different tipping elements, for different experiments: the coupled experiment with ECS =  $3.5^{\circ}$ C (black), the coupled experiment with ECS =  $3.5^{\circ}$ C and interaction strengths doubled (blue), and the coupled experiment with ECS =  $3.5^{\circ}$ C but excluding ASI and PERM (red). For a given element and scenario, the tipping risk is defined as the percentage of ensemble simulations, generated with different parameter sets, in which the element tips. For reference, the peak warming corresponding to a given level of emissions in the decoupled experiment is indicated on the upper axis of the plots.

only a limited destabilizing influence on AMOC, as its meltwater flux  $F_{GRIS}$  is in general small compared to the chosen critical threshold range ( $F_{GRIS}^+ \in [0.1, 0.5]$  sv), as shown in Figure D4.

Overall, these findings demonstrate the importance of including ASI and PERM in tipping cascade studies, even if they are not classified as tipping elements themselves. What matters most is their potentially nonlinear behaviour and, more critically, their strong coupling with other elements, rather than whether they strictly meet the definition of a tipping element.

#### D4 Risk of tipping events under current-policy pathways

Table D1 presents the risk of observing any tipping event for a scenario with cumulative emissions of 1200 PgC or less, across all experiments performed. We find that the experiments with the highest tipping risk are the ones with an ECS of  $5.0^{\circ}$ C, with a 95% risk for the coupled experiment and 88% for the decoupled experiment. On the other hand, the likelihood of observing cascades of size 3 or 4 is the greatest in the d=2 case, even surpassing that of the coupled experiment with ECS =  $5.0^{\circ}$ C.




Figure D4. Peak meltwater flux from Greenland as a function of cumulative emissions in the coupled experiment. How fast Greenland melts under a given scenario depends on the values of  $T_{+,GRIS}$  and  $\tau_{-,GRIS}$ , but also on the values of the coupling coefficients and the evolution of other elements, resulting in a large possible range for peak meltwater flux. Overall, the peak meltwater flux is quite small compared even to the smallest possible value for  $F_{GRIS}^+$ , resulting in a limited impact on AMOC.

This aligns with findings from Wunderling et al. (2021), who observed that interaction strength influences cascade distribution and size.

## 790 Appendix E: Increase in tipping risk as a function of emissions

In Figure E1, we examine how the increase in risk per additional 50 PgC of emissions varies as a function of cumulative emissions. In other words, we estimate the derivative of curves shown in Figure 6. Our aim is to compare our results to Möller et al. (2024). Using the model from Wunderling et al. (2021), they showed that the risk of observing tipping events by 2300 presents a nonlinear increase around 2 °C of peak warming, primarily driven by the increased risk of tipping the Amazon, a fast element.

In the coupled experiment, we find that the AMOC initially experiences the highest increase in tipping risk, with an increase of approximately 1%-2% per 50 PgC. We observe a large acceleration of the tipping risk (increase of increase) within 1000 PgC of cumulative emissions for AMAZ, WAIS, and EASB, but only the acceleration for AMAZ would be noticeable if we evaluated the tipping risk on short time scales. Comparison with the decoupled experiment shows that for AMAZ, the tipping risk acceleration for low emission scenarios primarily results from AMAZ starting to tip under these scenarios for some combinations of  $(T_+, \tau_-)$ , regardless of the interactions between elements and their feedbacks on temperature (see Figure 5). In contrast, the magnitude of the tipping risk acceleration for WAIS and EASB under low emissions scenario seems to be largely driven by interactions and feedbacks on temperature. An acceleration of the tipping risk within 1000 PgC is also observed for AMOC and GRIS, although with a smaller magnitude than for WAIS and AMAZ. These findings are consistent with the results from Möller et al. (2024).

**Figure E1.** Increase in tipping risk of the different tipping elements, per additional 50 PgC of cumulative emissions, and as a function of cumulative emissions. In other words, we plot the derivative of the tipping risk curves shown in Figure 6. The derivatives are estimated using a Savitzky-Golay filter with polynomial order 1 and a window length of 7 points. For reference, the peak warming corresponding to a given level of emissions in the decoupled experiment is indicated on the upper axis of the plots.

Table D1. Risk of observing tipping events for emissions scenarios with cumulative emissions below or equal to 1200 PgC, for different experiment settings. These experiments differ in several key aspects: the ECS used, whether interactions between elements are included, whether feedbacks of these elements on temperature are considered, and whether ASI and PERM are included. For each experiment and a given tipping event, the risk is computed here as the percentage of ensemble members with different parameters for which the tipping event occurs at least once in a scenario with cumulative emissions below or equal to 1200 PgC.

|                         |                  | Experiments                                     |            | any tipping | any tipping single tipping |        | destab | destabilising cascade              | scade  |        | stabilising |
|-------------------------|------------------|-------------------------------------------------|------------|-------------|----------------------------|--------|--------|------------------------------------|--------|--------|-------------|
| ECS                     | interacting TEs  | ECS interacting TEs feedbacks on GMT ASI & PERM | ASI & PERM | event       | event                      | size 2 | size 3 | size 2 size 3 size 4 size 5 size 6 | size 5 | size 6 | cascade     |
| 3.5 °C                  | no               | по                                              | yes        | 53.8        | 53.3                       | 6.0    | 0      | 0                                  | 0      | 0      | 0           |
| $3.5^{\circ}\mathrm{C}$ | yes              | yes                                             | yes        | 6.69        | 56.4                       | 19.1   | 4.2    | 0                                  | 0      | 0      | 2.4         |
| $3.5^{\circ}\mathrm{C}$ | yes              | ou                                              | yes        | 62.5        | 51.9                       | 13.8   | 2.9    | 0                                  | 0      | 0      | 1.9         |
| $3.5^{\circ}\mathrm{C}$ | ou               | yes                                             | yes        | 61.5        | 60.4                       | 1.9    | 0      | 0                                  | 0      | 0      | 0           |
| $5.0^{\circ}$ C         | ou               | ou                                              | yes        | 87.5        | 86.0                       | 3.8    | 0      | 0                                  | 0      | 0      | 0           |
| $5.0^{\circ}\mathrm{C}$ | yes              | yes                                             | yes        | 94.6        | 83.6                       | 38.3   | 10.5   | 9.0                                | 0      | 0      | 11.0        |
| $3.5^{\circ}\mathrm{C}$ | yes $(\times 2)$ | yes                                             | yes        | 9.62        | 42.8                       | 22.4   | 33.1   | 3.8                                | 0      | 0      | 4.4         |
| $3.5^{\circ}\mathrm{C}$ | yes              | yes                                             | ou         | 60.2        | 49.1                       | 15.6   | 2.4    | 0.1                                | 0      | 0      | 1.1         |



**Figure F1.** (a) Peak warming relative to 1850 CE as a function of cumulative emissions for the decoupled and coupled experiments. (b) Differences in peak warming between the coupled and decoupled experiments. Thick solid lines represent median values, while shaded areas show the full range across all simulations.

## Appendix F: Relationship between peak warming and cumulative emissions

In the decoupled experiment, the temperature evolution only depends on the emission scenario and is not impacted by the state of the tipping elements. The peak warming depends quasi-linearly on cumulative emissions (see Figure F1a). In the coupled experiment, the tipping elements impact the temperature evolution and thus the model may produce different temperatures under the same emission scenario, depending on the choice of parameters for the tipping elements. Nevertheless, the impact of elements is small and differences in peak warming between the decoupled and coupled setups under the same scenario and with the same parameter set do not exceed 0,5°C (Figure F1b). This is because the impact on temperature of the modelled tipping elements is overall relatively small in our model (see Tables 1 and 2) and also because the maximum impact occurs when elements are fully collapsed, which generally happens after peak warming.

In this paper, the tipping risk is always plotted as a function of cumulative emissions. In figures 6, 10, D3, and E1 the peak warming corresponding to a given level of emissions in the decoupled experiment (blue line in Figure F1) is indicated on the upper axis of the plots, in addition to the cumulative emissions on the lower axis. In Figures 7 and D1 (right panels), the tipping risk for the decoupled and coupled experiment is also plotted as a function of cumulative emissions, but only the peak warming corresponding to a given level of emissions in the decoupled experiment is given on the figure axis.

## 820 Appendix G: TIPMIP protocol

In the main text, we used SURFER as an exploratory tool to highlight important concepts and uncover robust phenomena, which could then be further investigated with more complex models.

Other applications are possible. For example, consider TIPMIP, an international, ongoing model inter-comparison project aimed at studying tipping dynamics in various Earth system components and assessing the uncertainties involved (Winkel-






mann et al., 2025). We could constrain SURFER's parameter ranges such to reproduce the results from TIPMIP. This calibration strategy would effectively transform SURFER into an emulator to conduct probabilistic risk assessments for specific scenarios. In this case, information flows downward within the model hierarchy, from complex, state-of-the-art models to reduced-complexity models.

Here, we showcase another possible use by running SURFER under the Tier 1 ESM TIPMIP protocol, providing a low-cost test of the protocol's feasibility, relevance, and potential challenges. This exercise offers preliminary insights into expected outputs and supplies a framework for interpreting them, serving as a proof of concept for TIPMIP.

The TIPMIP ESM protocol consists of a control run and 5 emission-based scenarios.  $CO_2$  is emitted at a constant rate such as to produce a warming of around  $0.2^{\circ}C$  per decade until warming reaches  $2^{\circ}C$  or  $4^{\circ}C$ , at which point emissions are zero for 50 years. Then emissions are either kept to zero until the end of the simulation, or set negative until warming is back to zero. For the  $4^{\circ}C$  warming case, there is also an additional scenario where negative emissions are introduced until temperatures fall back to  $2^{\circ}C$ , then kept at zero afterwards. Here we run SURFER under each of these scenarios and for each parameter set of our Monte-Carlo ensemble. We repeat this for 2 different model settings: the decoupled and coupled settings (see section 2.3). This constitutes 2 experiments, each consisting of  $5 \times 1000$  model runs. The decoupled experiment is equivalent to studying each tipping element separately in off-line models.

Figure G1 shows the evolution of the global mean temperature in SURFER across the five TIPMIP scenarios. In our model, a constant  $CO_2$  emission rate of  $\sim 13,3 PgC$  yr $^{-1}$  was required to obtain a warming rate of around  $0.2^{\circ}C$  per decade. It should be noted that the warming is not perfectly linear: reaching  $2^{\circ}C$  from  $0^{\circ}C$  requires about 90 years of emissions, while reaching  $4^{\circ}C$  from  $2^{\circ}C$  takes approximately 105 years. Additionally, temperature decreases during negative emissions are not symmetrical with the warming. For instance, while it takes 195 years of positive emissions to increase from  $0^{\circ}C$  to  $4^{\circ}C$ , only 167 years of negative emissions are needed to reverse this warming. This asymmetry is driven by the continuous action of natural land and ocean carbon sinks, and results in net cumulative emissions for the two scenarios with full temperature reversals: 120 PgC for the  $2^{\circ}C$  peak warming scenario and 373 PgC for the  $4^{\circ}C$  scenario. This carbon is accumulated in the ocean and land during the warming and cooling phases, and then partly released back to the atmosphere as the carbon reservoirs equilibrate after emissions are set to zero. This process explains the small rebound in atmospheric  $CO_2$  and temperatures observed in these scenarios.

Once emissions are set to zero, temperatures are fairly stable, with changes not exceeding  $0.5^{\circ}$ C until around year 500. At this point, temperatures decrease over multi-millennial timescales as atmospheric CO<sub>2</sub> gradually declines. In the coupled experiment, temperatures are up to  $0.56^{\circ}$ C higher than in the decoupled experiment due to feedbacks from tipping and other nonlinear elements. These feedbacks may affect tipping risks and should be considered when defining scenarios. Specifically, it is important to specify whether the feedbacks of studied elements on temperature, especially faster ones, are included when determining the constant emission rate that results in a  $\sim 0.2^{\circ}$ C warming per decade.

Figure G2 shows the decoupled tipping thresholds for the 5 TIPMIP scenarios in the  $(T_+, \tau_-)$  space, similar to Figure 5 in the main text. The tipping risk for each element is shown in Figure G3 for both the decoupled and coupled experiments. To

**Figure G1.** Temperature evolution for the different scenarios of the TIPMIP Tier 1 ESM protocol. Plain lines indicate the temperature evolution in our decoupled experiment settings, that is not including the interactions between tipping elements and their feedback on temperature. The dotted lines and shaded areas indicate the median and maximum range of temperatures in the coupled experiment settings, respectively.

Figure G2. Tipping behaviour thresholds for the tipping elements, as a function of their critical temperature  $T_+$  and their intrinsic time scale  $\tau_-$ . The gray rectangles indicate the range of possible parameter values for each element. The plain coloured lines represent the tipping thresholds for the different scenarios of the TIPMIP Tier 1 ESM protocol, and in our decoupled experiment (interactions between tipping elements and their feedback on temperature are not included). An element with a  $(T_+, \tau_-)$  combination that is below a given threshold will tip for this scenario. The coloured dotted lines are the theoretical predictions for the tipping thresholds based on the criterion from Ritchie et al. (2019), see equation 38.

account for time-limited simulations from some Earth system models used in the TIPMIP project, we also present tipping risks

based on only the first 500 years of the model runs, identifying whether an element is at risk of tipping within this initial period.

For AMAZ and AMOC, tipping risk remains relatively consistent whether we consider only the first 500 years of the simulations or the full run, as these are fast elements. Additionally, scenarios with a peak temperature of 4°C consistently show


**Figure G3.** Tipping risk of the tipping elements, for the different scenarios of the TIPMIP Tier 1 ESM protocol, and for different experiment settings (baseline, coupled, only interactions, only feedbacks). For a given element and scenario, the tipping risk is defined as the percentage of ensemble simulations, generated with different parameter sets, in which the element tips.

higher tipping risk than those with a 2°C peak. Among scenarios with the same peak temperature, those with higher cumulative emissions (i.e., less negative emissions) present an increased tipping risk.

For GRIS, WAIS, and EASB, the tipping risk is zero when considering only the first 500 years of the simulations, reflecting their nature as slow-tipping elements that require longer timescales for meaningful study. The tipping risk increases across the scenarios with cumulative emissions, regardless of the peak temperature reached. This is because, for slow-tipping ice sheets, long-term temperatures, driven primarily by cumulative emissions, are more influential than short-term temperature fluctuations. Long-term temperature evolution also depends on each model's representation of the carbon cycle, which can yield significantly different long-term CO<sub>2</sub> concentrations even for identical emission scenarios (Archer et al., 2009; Kaufhold et al., 2025). Consequently, if emissions-based protocols are used to study the ice-sheets (for example in EMICs), whether they will ultimately tip in a given model may depend more on its simulation of long-term CO<sub>2</sub> draw down than on its specific representation of the ice sheets themselves (Couplet et al., 2025). In TIPMIP, domain-specific protocols are planned for the ice-sheets to address the longer time scales involved.


Author contributions. VC, and MC conceptualised the project. VC, and MC contributed to the methodology by developing the model. MC managed and supervised the project. VC wrote the model code and carried out the model simulations. VC did the visualisations. VC wrote the original draft. MC reviewed and edited the paper.

Competing interests. MC is a member of the editorial board of ESD.

Acknowledgements. We thank Marina Martinez Montero for helpful discussions and general support in developing the SURFER model. We acknowledge the Global Carbon Project, which is responsible for the Global Carbon Budget and we thank the different modelling groups for producing and making available their model output. The scientific colormap lipari (Crameri, 2023) is used in most figures. The authors used a language model (ChatGPT, OpenAI) to assist with minor text editing and phrasing improvements.

Financial support. Most of this work was done when Victor Couplet was in UCLouvain, supported by the Belgian National Fund of Scientific Research (F.S.R-FNRS) under the Aspirant Fellowship FC 38941, and by the "Fonds Speciaux de Recherche" (FSR) of UCLouvain. Publication costs are covered by the FSR.

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
