# Peer review of "Tipping interactions and cascades on multimillennial time scales in a model of reduced complexity"

_EGUsphere, 2025_

## Author Comment (AC1)

**Author response to Referee comments**

**January 2026**

We sincerely thank both reviewers for taking the time to thoroughly review our manuscript and for their constructive comments and suggestions. We are pleased that both reviewers found the paper to be generally well written and well structured. A central concern raised by both reviewers relates to our use of uniform distributions for sampling a large number of parameters, including how this choice can be justified and what implications it has for the interpretation of our results and the limitations of our study. We therefore address this issue first in a general manner before turning to the more specific comments.

We chose uniform distributions for parameter sampling as a pragmatic and transparent approach to exploring the broad and substantial uncertainties associated with tipping thresholds, interaction strengths, and intrinsic timescales. In many cases, the available literature provides only minimum, maximum, and best estimates, which themselves are subject to considerable uncertainty and expert judgment. Given this situation, uniform sampling allows us to avoid imposing additional assumptions about underlying probability distributions that are not well constrained by the data.

We acknowledge that using uniform distributions may over- or under-represent certain regions of parameter space relative to expert-informed distributions. However, even expert-informed distributions would not guarantee real-world applicability, given the simplicity of our model. Indeed, a proper Bayesian analysis suitable for decision-making—such as setting an insurance premium or planning infrastructure—would require not only carefully designed priors, typically based on expert elicitation, but also a strategy for addressing structural uncertainties, including model errors. This is a complex undertaking that lies well beyond the scope of the present study.

Our aim is therefore not to provide a probabilistic risk assessment in a strict real-world sense. Instead, our goal is more modest: to extract robust, structural properties of the climate system, such as the existence and structure of tipping cascades, their typical initiators, and their dependence on interactions and feedbacks. For instance, in contrast to previous studies, we identify the AMOC as the primary potential initiator of cascades. This, we believe, is an advance to current wisdom. Before reaching this conclusion, we ensure that it is robust across a wide range of model parameters. Uniform sampling offers a convenient strategy for this purpose, as it is easy to implement with a fast model and allows broad coverage of a multidimensional parameter space, capturing potential nonlinearities or unexpected behavior that might be overlooked in a more limited sensitivity analysis.

In summary, we confirm that our objective is to assess robustness rather than to perform

a quantitative analysis of posterior distributions. We will clarify this objective and the associated limitations in the revised manuscript.

**Comments from Referee 1**

The manuscript Tipping interactions and cascades on multimillennial time scales in a model of reduced complexity evaluates the tipping risk for six interconnected tipping elements of the climate system for a range of emission scenarios. Specifically, the authors consider how the tipping risk changes when including coupling between elements and feedbacks to the climate. The authors show that interactions between tipping elements generally increase the risk of tipping via cascades that can unfold over multi-millennial timescales. This result is consistent with previous studies; however, the authors also find the AMOC to be the main initiator of tipping cascades as opposed to the Greenland ice sheet. The authors attribute this to assuming a flux (derivative) coupling between the Greenland ice sheet and the AMOC instead of a coupling that is proportional to the remaining ice sheet. The manuscript is generally well written and easy to follow, but I have some general and specific comments that I would like to see the authors address before supporting publication.

**General comments:**

Further clarification on the classification of cascades is required. For instance, in Figure 9 there appears to be more than 1,000 cumulative counts (the number of different parameter setups) of destabilising GRIS events. Presumably this is because of stabilisation cascades and so the same parameter setup can be counted multiple times. For example, for a single parameter setup that must be scenarios where GRIS tips for the first time at scenario $j$, then stabilises at scenario $k$ (i.e. due to AMOC tipping), and then tips again in scenario $l$ for $j < k < l$. Is this indeed the case? Similarly, for the same parameter setup, if only GRIS changed tipping status at scenario $j$ and only WAIS changed tipping status at scenario $k$ for $j \neq k$, then these would be represented by a single red dot for GRIS (first bar) and a single red dot for WAIS (third bar)?

Yes, this is correct. We will clarify that multiple tipping events can be counted for different scenarios for the same parameter setup.

It appears counter-intuitive that not every (in particular) stabilising cascade has an initiator (Figure 11). As suggested by L530–531, for there to be no initiator in a stabilising cascade there must be a change in the permafrost or sea ice that causes an element such as the AMOC to tip, which then stabilises GRIS. If correct, this needs to be highlighted more clearly.

This is also correct and will be highlighted more clearly.

For the Latin Hypercube Sampling, the authors employ a uniform distribution for all parameters, but with little apparent justification. For the critical temperature thresholds and transition timescales, not only are the minimum and maximum estimates provided, but also a best estimate, which appears to get overlooked. For example, the EASB has an estimated temperature threshold between 2°C and 6°C, but a best estimate at 3°C, so arguably more weight should be applied to lower threshold values. By using a uniform

distribution, the tipping risk would therefore be underestimated. Further, it is not a linear transform between the transition timescale and $\tau_-$, which has greater sensitivity for low values, so further justification needs to be given for choosing a uniform distribution over $\tau_-$ as opposed to the transition timescale.

See general comment on using uniform distributions. Furthermore, we choose a uniform distribution over $\tau_-$ as opposed to the transition timescale because that is the model parameter that we can directly sample. An alternative would be to sample the critical temperature and transition timescale for each element and then compute separately the right $\tau_-$ value needed for that transition timescale. But then, on top of extra calculations, we would run into the problem that the transition timescale also depends on the forcing scenario. Another alternative would be to follow the same procedure as we do currently but find the distribution of $\tau_-$ such that the distribution of transition timescales for a given scenario (Figure C1) is uniform. However this is much more complex to do.

**Specific comments:**

L87: "... neither exhibits signs of bistability,..." — do not show signs of bistability in an ESM or the conceptual model? Also, explicitly state again here that this is for Arctic sea ice and boreal permafrost.

To our knowledge, Arctic sea ice and boreal permafrost do not show signs of bistability in more complex models, and so we have chosen to represent them as monostable elements in our conceptual model.

L95–100: Is the sea level rise just an output of the model, or does $S_{\mathrm{gl}}$ explicitly feed into the tipping dynamics somewhere that I have missed? I.e. what is the motivation for including it here? Maybe nearer the end of the section, a small comment stating that sea level rise can be determined, with a reference to the model paper, would be sufficient. Additionally, why is there no contribution from ice sheets and only the mountain glaciers (also noted on L75)? If ice sheets are included in mountain glaciers, then this is confusing, with mountain glaciers typically being treated as their own tipping element.

Sea level rise is an output of the model and is computed as the sum of six contributions: mountain glaciers, the Greenland Ice Sheet, the West Antarctic Ice Sheet, the East Antarctic Ice Sheet, East Antarctic subglacial basins, and thermal expansion. The evolution of mountain glaciers is directly simulated using their sea-level-rise contribution as a diagnostic variable ($S_{\mathrm{gl}}$), whereas for the ice sheets we simulate ice volume , which is subsequently used to compute their respective sea-level-rise contributions. We will modify the relevant sentences to avoid all possible conclusion.

L180: The parameter $x_+$ is also fixed...?

Yes indeed. We will correct L180 accordingly.

L186–187: Presumably SSP1-2.6 does not extend until year 100,000 CE, so what happens to methane emissions at the end of SSP1-2.6? Are they assumed to be zero?

Yes, methane emissions follow the extended SSP1-2.6 emission scenario until 2300, then are assumed to be zero. This is stated in Appendix A so we didn't specify it in the main text.

L210–211: The authors claim that the choice of $T_-$ does not affect the results of the "forward tipping points". However, as the authors previously state, specifying $T_-$ determines $x_-$, which determines all the coefficients (Eqs. 5–8) in Eq. 3. Therefore, does this not affect characteristics such as the curvature of the fold at $T_+$ and thus the overshoot behaviour?

Yes, the reviewer is correct that the choice of $T_-$ determines $x_-$ and all associated coefficients (Eqs. 5–8) in Eq. 3. Consequently, it also affects characteristics such as the curvature of the fold at $T_+$ and thus the overshoot behaviour. Had we chosen different fixed values for $T_-$, we would indeed have obtained different numerical results. However, since we calibrate the sampling range for $\tau_-$ after fixing $T_-$, this would likely compensate for part of the change, and the results would therefore remain similar, at least qualitatively. Nevertheless, we agree with the reviewer that our previous claim that the choice of $T_-$ does not affect the results is inaccurate, and we will remove it.

L222 & L227: Where do the values for $\tau_+$ come from, and what transition timescales do these correspond to? Arguably these are less important than $T_-$ and $x_-$, i.e. the results seem less dependent on the choice of $\tau_+$?

The values chosen for $\tau_+$ were based on experimentation with SURFER during model development and on attempts to loosely fit a range of results reported in the literature. Admittedly, these choices are to a large extent arbitrary, or, more charitably to the authors, may be described as "educated guesses". The parameters $\tau_-$ and $\tau_+$ do not correspond directly to transition timescales, but rather to the intrinsic timescales of the associated elements when their state variable decreases ($\tau_-$) or increases ($\tau_+$). For tipping elements, these intrinsic timescales largely determine the transition times from the upper equilibrium branch to the lower equilibrium branch ($\tau_-$), or vice versa ($\tau_+$). Since our analysis primarily focuses on forward tipping, i.e. transitions from the upper to the lower equilibrium branch, the results depend mainly on the choice of $\tau_-$ and are much less sensitive to the choice of $\tau_+$. We will add a sentence to provide a bit more information on the choice of values for $\tau_+$.

L397 & L399: "below" and "above" appear to be the wrong way round?

No, the sentences are correct. Confusion may arise because of our terminology of "tipping threshold" (see next comment). Reviewer 2 suggested to rather use "boundary curve separating safe and unsafe overshoot". In this case, the area below the boundary curve is unsafe (leads to tipping) and the the area above is safe. We will adapt of terminology to avoid confusion.

L404–405: Many readers will associate the "tipping threshold" with the critical global warming threshold rather than the threshold that separates tipping from not tipping. A comment emphasising this distinction is therefore important.

As also suggested by Reviewer 2, we will adapt our terminology (see comment above). Thanks for pointing out this source of confusion.

L407–408: Specifically, what assumptions?

Under the assumption that the forcing change is slow enough compared to dynamics of the element. We will state this explicitly.

L504–505: However, the peak temperature can still be above 2.7°C, though after 2100? Please also note the revised figure (according to Climate Action Tracker, 2025) is 2.6°C.

Yes, the peak temperature can still exceed 2.7°C after 2100. Thank you for pointing out the revised figure. We will either redo the analysis using the updated estimate or, more likely, explicitly state that the 2.7°C value corresponds to the estimate available in October 2025.

L509: Maybe use "probability" instead of "risk" when referring to a stabilising cascade.

We would prefer to keep the terminology consistent and use "risk" also for stabilising cascades. The "risk" we define should be interpreted as occurrence frequencies or relative likelihoods within the model framework rather than direct estimates of real-world tipping probabilities, which is why we want to avoid using "probability". See also general comment on uniform distributions.

L600: As previously mentioned, arguably the choice of $T_-$ and the corresponding value for $x_-$ already matters for "forward tipping points" and not just "back tipping" points.

Yes indeed. We will modify the sentence accordingly.

L650: Presumably, the first term in the denominator should be $F(2024)$ to give $E(t) = E(2024)$ when $t = 2024$ in Equation (A1)?

Yes, thanks for spotting this mistake.

**Technical comments:**

L40: "several thousand of years" → "several thousands of years"

L51: "from Amazon" → "from the Amazon"

L92: "greenhouse gas." → "greenhouse gas emissions."

L216: Add a space between "$\tau_+$" and "in"

Figs. 5 & G2 legends: "Rictchie" → "Ritchie"

L566: "(Deutloff et al., 2025)" → "Deutloff et al. (2025)"

L575: "imitator" → "initiator"

L593: "he" → "the"

L797: please correct "increase of increase"

L812: "0,5°C" → "0.5°C"

L841: "13,3" → "13.3"

Thanks for outlining these typos, we will fix them in the revised manuscript.

**Comments from Referee 2**

This study uses SURFER, a low-complexity ESM with six tipping elements and two non-linear elements as well as important climate feedbacks, to assess the risk of tipping cascades. It is well structured and largely well-written. Within the limits of this framework, the analysis is carefully executed and has clear didactic value, particularly in demonstrating transient tipping behaviour under overshoot scenarios. Overall, this publication is a next logical step within a series of simple network models of tipping elements by adding transient feedbacks to the climate system, such that I generally support its publication.

However, a major concern is that the simplified risk metric based on Monte Carlo occurrence counts, combined with large uncertainties in tipping thresholds and highly idealised interactions, makes it difficult to draw conclusions about real-world risk and may lead to over- or understatement when taken out of context. This is a common limitation of these simple models (beyond this study) but should be addressed more transparently by elaborating more on how to interpret the results, and which conclusions about the real world can not be done, or only in limited fashion. In particular when stating that risk may be overestimated, it is on the authors to provide more reflections on how the limitations of this approach also limits the extent to which such statements can be done. Especially the limitations of the expert-based parametrisation of the interactions should be expanded on. Apart from this, there is a range of minor and major specific comments below.

See general comment on uniform distributions. Regarding the parameterisation of interactions, we already provide a discussion in Appendix D2. Inferring precise numerical values from expert judgment is inherently challenging, which is why we perform a sensitivity analysis. In the revised manuscript, we can expand on this point a bit more in the main text—for example, in the Discussion section—to highlight the rationale and limitations of our approach.

**Specific comments**

**Overall**

- Highlight novelty of the study more clearly, and its limitations
- Provide summary figure(s) and table(s) that help the reader dissect the different parts of the paper better (see comments below)
- Revise section 2 to make sure all expressions and terms are introduced in the right order (in the current version the reader needs to scroll back and forth to make sense of this).
- Make sure that the figure and table captions contain all necessary information to read them. If there are too many acronyms or terms that don't fit the caption, refer to one of the glossary tables requested above
- Adapt figure sizes, especially font size, to make them more readable.

Thank you for these overall suggestions, which will definitely improve the quality of the paper. We will take them into account in the revised manuscript.

**Abstract**

- Introduce with a sentence explaining what tipping is. Also, "chain reaction could lead to substantial and possibly irreversible changes in the Earth's system" → this is already true for a single tipping element.

  Yes, of course tipping cascades can lead to even more substantial changes but this is already true for a single tipping element. We will adapt the sentence to reflect that nuance. We will also add a sentence to explain what a tipping element is.

- From the abstract, the novelty of the study is not clear. It is certainly building confidence that it is consistent with previous studies, but it is not clear how this study differs from previous ones in its approach.

  The novelty of this study lies in its investigation of tipping cascades within a single, unified framework that combines direct interactions between tipping elements, indirect interactions mediated by feedbacks on the global mean temperature, and non-stationary temperature forcings. We will make this clearer in the abstract.

**Introduction**

- Please overall check that tipping-related terms are carefully defined. E.g. distinguish climate tipping points (Lenton et al 2008; Armstrong McKay et al 2022) from Earth system tipping points (Lenton et al, 2023); also tipping elements and tipping systems. Cite IPCC where appropriate (e.g. at "may be abrupt and/or irreversible")

  We will make sure that tipping related terms are carefully defined in the revised manuscript. However, as we understand it , climate tipping points and Earth system tipping points are synonyms, same as tipping element and tipping system.

- On critical slowing down / early warning signals: Please group the references by system, and also include recent literature. For the different systems, criticism of the (in parts very simplified) EWS analyses has been published in the last years – to my understanding not fully refuting EWS, but justifying that statements on critical slowing down should be done in a weaker way (e.g. "...exhibit signals that are consistent with critical slowing down, which is however debated...")

  We organise references by system and look for recent literature. We have already included a reference that criticizes parts of EWS analysis (Ben-Yami et al., 2024), but will try to cite others.

- I stumbled over interactions and feedbacks, and had to go back to recent reviews (e.g. Wunderling et al 2024) to confirm that "interactions between tipping elements" indeed does only include direct influences (e.g. via precip pattern changes, FW release, regional circulation changes etc.). A reader (myself included) might think that the Amazon "interacts" with the ice sheets too via release of carbon (and subsequent global warming) when it tips. It gets clearer when reading on, that the latter kind of "interaction" is labelled climate feedback here, and is not included in "interactions between tipping elements". Maybe it would help to make this more clear in the beginning, by explicitly stating that tipping elements can interact directly (e.g. freshwater discharge from GRIS destabilises the AMOC) and can impact each other indirectly, via modification of the global climate, also providing examples for each.

> Thanks for pointing out this possible source of confusion. We will make the distinction between direct and indirect interactions clearer in the beginning.

- The summary of recent work using simple climate models in this context was well done. I believe the paper could greatly benefit from a conceptual overview figure here, including

  - A map of all considered tipping elements, sketching their interactions and feedback to the global climate

  - A global temperature curve as in figure 4a, conceptually sketching a "prescribed" overshoot (as in Wunderling et al 2023, Möller et al 2024) vs one that includes the temperature feedbacks on the global climate, e.g. an overshoot curve that includes the warming induced by an AMAZ collapse

> Thanks for this suggestion. We will add such a figure.

**Methods**

SURFER model

- Although brevity is appreciated, some more background on SURFER would be good in the beginning; at least statements on how it performs against observations and CMIP (i.e. some more context for "reliably estimates .... in response to anthropogenic GHG emissions")

> We will provide a bit more context on SURFER in the revised manuscript.

Representation of tipping elements

- There are many symbols defined somewhere in the text (sometimes not directly around the corresponding equation). Readers would greatly benefit from a table in the appendix with all introduced symbols. Please revise section 2.2, making sure that all the terms are introduced in a logical order and explained (e.g. $T_U$ in eq 1, which is only later introduced in line 152). Same with equation 16, where a sum over $j$ and $\delta L_j$ are introduced but only explained quite a bit later.

> We will add a summary table listing all symbols, thank you for the suggestion. We will also revise Section 2 to improve the logical order in which new terms are introduced.

- Not critical, but $\delta q$ usually suggests a small deviation from a state (perturbation), but in this context it is meant to be the anomaly wrt to a preindustrial state, which is "large" in the sense that it is not a perturbation to a system but fundamentally changes the stability landscape. So $\Delta q$ would be more appropriate, reserving $\delta q$ for small variations e.g. for EWS / resilience studies. Also, there is some inconsistency, as most expressions involve $\delta q$, but e.g in line 118 and in equation 3, there is also $q$ (without $\delta$). And then $q_{max}$ in eq 35 should also be a $\Delta q_{max}$.

> We understand the reviewers concern, however we would like to keep using $\delta q$ to be consistent with other symbols introduced in previous SURFER versions, eg

$\delta T_U$ which represents the global mean temperature anomaly compared to the pre-industrial period. We will check for inconsistencies in notation and will correct them.

- It is not clear to me if the inclusion of tipping elements into SURFER is an addition or a modification. E.g. a "non-tipping" SURFER version includes land (Fig 1) – is that the total global carbon $X$ stored in non-marine vegetation and soil? If yes, how is the AMAZ represented in the modified model? Is then $X = X_{\text{linear}} + X_{\text{AMAZ}}$? I.e. it would be enlightening to see what the state variables in the different SURFER versions are, and if the inclusion of tipping elements modifies the differential equations and/or adds additional state variables.

  SURFER v3.0 already included the Greenland ice sheet and Antarctic ice sheet. In this new version, the Antarctic ice sheet is split between different components (WAIS,EAIS, and EASB) and we add parametrisations for the AMOC, the Amazon Rainforest, the boreal permafrost and the Arctic sea ice. The land carbon reservoir $M_L$ is the total carbon stored in non-marine vegetation and soil, including Amazon Rainforest and permafrost. We will clarify this in the revised version.

- When discussing the stability structure, readers might benefit from additional links to literature such as textbooks by Dijkstra, Strogatz etc.

  We will add these references.

- Eq 4: How does $H_i$ change..?

  $H_i$ changes as a function of $x_i$, see Eq 3. We will add a definition of $H_i$ outside of Eq 3 to improve readibility.

- On the comment in line 155ff: The critical temperature thresholds for tipping elements are (or intend to be) already accounting for "all" climate-related effects that might lead to tipping. Shouldn't the global warming level already be a good proxy for all other climate-related forcings, especially different ocean temperatures? If you want to make the case that there are other forcing influences, then wouldn't it be appropriate to also list other systems such as land use for AMAZ?

  The point here is that, although the critical temperature thresholds for tipping elements are intended to account for the full range of climate-related influences that may lead to tipping, this assumption may not hold equally well for all elements. In particular, the AMOC may respond not only to changes in surface ocean temperature but also to variations in ocean stratification, which cannot be related to surface temperature in a simple or unique way. Indeed, for the same surface temperature, ocean stratification can differ depending on the rate of warming. We agree that it would also be appropriate to list land-use change as a relevant driver for the Amazon rainforest, and we will include this in the revised manuscript.

- Related small caveat: If intermediate and deep ocean temperatures are not used in this work, please refrain from introducing more terms (bracket in line 157), especially since the I subscript is already used above for the inflexion point

  Indeed, we will remove the bracket in line 157.

- Eq 16: Why is it needed to separate $\epsilon_{ij}$ from $\delta L_j$ rather than just having one $\delta L_{ij}$ matrix? Eq 17: why is $q$ included in the forcing? I mean, why is the second term in the forcing not just a function of $\delta L_{ij}$?

  Both of these choices seemed to us the most elegant way to formalise the framework, although we agree that this is ultimately a matter of personal preference. We separated the $\epsilon_{ij}$ terms from $\delta L_j$ from the outset because they naturally arise when defining the interaction strengths (i.e. $\delta L_{ij} = \epsilon_{ij}\delta L_j q_{*,i}$). We factored out $q_{*,i}$ because we felt that this led to a cleaner presentation of the equations, but this too is largely a matter of stylistic preference.

- **Table 1**: In the caption, please add explainers for $T_i$ etc

  We will add explainers for the parameters that determine feedbacks of elements on GMT (last three columns).

- Overall, it would make more sense to me to have the calibration part directly after section 2.2, and have the experimental setup last in section 2 (i.e. switch the order of sections 2.3 and 2.4)

- Up for the authors to decide (might be a matter of taste): To me, equations 3,4,16,17 are the important ones in section 2.2, and this section might be easier to read if it focussed on these (and expanded the explanation according to the other comments made here). Eqs 5-15 "only" describe how the parameters are inferred from data/literature, and could be deferred to the calibration section.

  We will revise section 2 to improve flow and readibility. We will consider switching the order of section 2.3 and 2.4, and moving Eqs 5-15 to the calibration section.

Experimental Setup

- What is the reason for 100,000 years of simulation?

  The main reason for choosing a simulation length of 100,000 years is that ice sheets can take very long times to transition to a new equilibrium, particularly when a tipping point is only marginally crossed, as the dynamics near the unstable manifold are inherently slow. In addition, the numerical integration scheme employs an adaptive time step, which allows for very large time steps once the system approaches equilibrium. As a result, simulations of 100,000 years are not substantially more computationally expensive than simulations of say 50,000 years, and 100,000 years appeared to be a convenient round number. We did not extend the simulations beyond this timescale because, on longer timescales, astronomical forcing, currently not included in SURFER, would become a dominant driver of the system's evolution.

- The $CO_2$ and $CH_4$ scenarios don't seem too compatible, what is the reason to consider e.g. 5000PgC together with a SSP1-2.6 $CH_4$ scenario?

  For simplicity, we chose to focus exclusively on CO2 emission scenarios. However, in order to better reproduce historical warming, it was necessary to include historical CH4 emissions. We therefore prescribed a single methane emission scenario that is used consistently across all CO2 emission scenarios. Rather than assuming zero

CH4 emissions after 2024, which would introduce a sharp discontinuity in emissions, we adopted an SSP scenario. We selected SSP1-2.6 in order to remain on the conservative side of warming.

- Latin hypercube sampling appears to be the standard for these kind of studies. One could argue that with the amount of information that is available on critical thresholds, any sampling is as wrong as another. However, Armstrong McKay et al. provide expert-based best estimates for the critical threshold, which could be a good starting point for a sensitivity analysis that draws the MC samples not evenly from the full range but rather, say, from a normal distribution centered on that best estimate. Another option would be to do a conservative approach and only draw samples from the minimum to the central estimate, i.e. the lower range of the Armstrong McKay et al's uncertainty ranges. It would be of great added value to see if an alternative sampling would lead to a qualitatively different result.

  See general comment on uniform distributions. We will try to repeat the coupled experiment with an alternative sampling to see how this affects the results. We can add this in appendix D.

Calibration

- Table 2: What is the justification for deviating from Armstrong McKay et al. (2022) for the EAIS timescales. A reference or more elaborated reason than "This provides a better estimation of committed sea level rise" is needed here.

  During the development and testing phase of SURFER, when using very long time scales for the regrowth of the EAIS, we saw that the model would overestimate Antarctic sea level rise contribution compared to other models, which is why we have chosen shorter time scales than Armstrong McKay et al. (2022). We will elaborate a bit more on this in the revised manuscript.

- Line 209 and 222: If the focus is primarily on "forward tipping points", why are different "backwards-tipping" timescales included?

  The parameters $\tau_-$ and $\tau_+$ do not correspond directly to transition timescales, but rather to the intrinsic timescales of the associated elements when their state variable decreases $(\tau_-)$ or increases $(\tau_+)$. For tipping elements, these intrinsic timescales largely determine the transition times from the upper equilibrium branch to the lower equilibrium branch $(\tau_-)$, or vice versa $(\tau_+)$. But they also govern the evolution of the elements even in cases where no tipping occurs. In all scenarios, global temperature decreases on multi-centennial to millennial timescales after the end of emissions due to the action of natural carbon sinks. As a consequence, the state variables of many elements will tend to increase regardless of whether they have tipped, making it important to explicitly include the timescale $\tau_+$.

- Equation 19 and 20: The numbers below the fractional line clearly indicate a unit conversion. However these should be included in the units of the quantities used. This would also make the formulae easier to understand.

  We agree, and will add the units in the revised manuscript.

**Results**

- Next to the definition of "risks" please add a disclaimer of how to read "likelihoods" in the remainder of the paper

"Likelihoods" should be read as occurrence frequencies or relative likelihoods within the model framework rather than direct estimates of real-world tipping probabilities. See also general comment on uniform distributions.

Individual tipping elements

- Is there some intuition behind $4a_0^2\kappa$? It seems to link somewhat to the slope of the stability landscape close to the critical point, but it would be nice to have some explanation

This comes from Ritchie et al 2019, and will look up the reference to see if we can find an intuitive explanation.

- Similarly, is there an explanation why $\tau_-$ plays a role in eq 38, but not $\tau_+$?

During an overshoot, $H_i$ is negative (the state variable $x_i$ of element $i$ decreases), and so only $\tau_-$ plays a role in determining if the overshoot is large and long enough to cause tipping. We will add a note on that in the revised manuscript.

- Figure 4: include $T_{\max}$. $t_{\text{over}}$ ... in the figure

Will do, thanks for the suggestion.

- Confusing language: In the context of tipping points, transgression of a threshold == tipping. In lines 392ff "threshold" is used in the context of eq 38, and suddenly being above the threshold == no tipping. Since that threads through the results chapter, I strongly suggest to find an alternative terminology. Actually Ritchie et al 2019 refer to this as the "boundary curve separating safe and unsafe overshoot", which seems way more appropriate.

This is indeed confusing and was also pointed out by reviewer 1. We will adapt our terminology, thanks for the suggestion.

- Figures 4 and 5 are very nice illustrations of the "Ritchie theory" and the consequences of coupling on safe overshoots. Since (to me) they are good takeaways of the paper, I suggest revising them to make sure they can serve as standalone explainers. E.g. rather than labelling 2000PgC, $T_{\max} = 4.2°C$ in 5b, explicitly label the boundary curve as "theoretical boundary for [overshoot scenario]" and hash out one side of the boundary with the label "unsafe overshoot", the other one with "safe overshoot". For the [overshoot scenario] either use the 2000PgC overshoot trajectory reaching $T_{\max} = 4.2°C$, or put that info in the caption and in the (shareable) figure rather put the peak temperature, landing temperature and overshoot duration.

Thank you for these suggestions. We will revise figures 4 and 5 based on them. Note that in figure 5b, it is the experimental/effective boundary, not the theoretical one.

- Line 400 → refer to Figure 5 (otherwise one might try and find this somewhere in Fig 4..)

We will refer to Figure 5.

- Line 411 → Please add what landing climate reaching 2.9°C with 1000PgC correspond to and which overshoot duration, to make clear that 2.9°C and higher are only "safe" when the overshoot is short and has a low landing climate

  We will add information on overshoot duration and landing climate, thanks for the suggestion. Note that in the case of Greenland, overshoot can still be very high and long (on a human time scale) and still be safe (see example Fig 4).

- Line 426 → simplify sentence; "...inevitable without AMOC influence, which can reduce the aggregated tipping risk when it has a stabilising effect."

  We will simplify the sentence.

- Line 441 → baseline should be called "decoupled"

  We will correct this and other instances in the text where "baseline" still appears (we changed terminology in the late stages of the manuscript prepartion).

Tipping cascades

- I think it would greatly improve the understanding of the results if the definition of a tipping cascade was explained more thoroughly early on. The brief "technical" definition in line 456ff only somewhat starts making sense with the thoughts introduced only later in the discussion in lines 581ff.

  Thanks for the suggestion. We will move lines 581-589 from the discussion to just after the definition of tipping cascades.

- Figure 9: Interesting representation of the results. Some comments on readability:
  - Write somewhere what the horizontal axis is supposed to be (the different combinations of tipping events)
  - Maybe visually rework the inset table to make more clear how to read it. E.g. "Single tipping events" is formatted exactly like "tipping events" and "percentage of..." and could be mistaken for a heading.

- Figure 10: Add "from 2024" to the caption

  Will be added.

- Lines 526ff: I don't quite understand "This can be verified by rerunning the model ...". Does that mean that if you want to know if AMOC is a potential initiator of a cascade occurring in run $j$, you compare run $j$ to run $j-1$ in the uncoupled case? Or how should I understand "This can be verified by rerunning the model ..."? What does "with all other elements ... held at their values from scenario $j-1$.." mean?

  To determine whether the AMOC is a potential initiator of a cascade occurring in run $j$, we rerun the model for scenario $j$ in the coupled setup, but with the state evolution of all other elements ($x_i$, $i \neq$ AMOC) prescribed to follow their trajectories from run $j-1$ rather than evolving dynamically. This approach allows us to isolate

the "direct" effect of the additional 50 PgC of $CO_2$ emissions on the AMOC from the "indirect" effects mediated through changes in the other elements. We will try to make this clearer in the revised manuscript.

**Discussion + Conclusion**

I'd wish for a more critical assessment on the limitations of this study, and overall these kinds of studies (including Wunderling et al. etc). The (valid) points brought up mostly address the limitations from a technical side – calibration, bifurcation- vs. other types of tipping etc. But more fundamentally,

- What are the epistemological limitations?

- The claim tipping risk might be overestimated when just considering critical thresholds makes sense within the limitations of this study.

- What is the merit of having 800000 model runs, if they're based on uniform sampling of very broad uncertainty ranges for parameters?

Overall, more reflection here (and acknowledgement of the limitations) would provide the reader more confidence that the results should not be taken at face value in a risk assessment kind of way. Additionally, the limitation of the study with regards to the expert based calibration of most of the interactions should be highlighted.

See general comment at the beginning of this repsonse.